

# Lacing topological orders in two dimensions:
# Exactly solvable models for Kitaev's sixteen-fold way

**Jin-Tao Jin[1], Jian-Jian Miao[2⋆] and Yi Zhou[3,4,1,5†]**

**1** Kavli Institute for Theoretical Sciences, University of Chinese Academy of Sciences,
Beijing 100190, China.
**2** Department of Physics, The Chinese University of Hong Kong,
Shatin, New Territories, Hong Kong, China.
**3** Institute of Physics, Chinese Academy of Sciences, Beijing 100190, China.
**4** Songshan Lake Materials Laboratory, Dongguan, Guangdong 523808, China.
**5** CAS Center for Excellence in Topological Quantum Computation,
University of Chinese Academy of Sciences, Beijing 100190, China.

⋆ jjmiao@phy.cuhk.edu.hk , † yizhou@iphy.ac.cn

## Abstract

A family of two-dimensional (2D) spin-1/2 models have been constructed to realize Kitaev's sixteen-fold way of anyon theories. Defining a one-dimensional (1D) path through all the lattice sites, and performing the Jordan-Wigner transformation with the help of the 1D path, we find that such a spin-1/2 model is equivalent to a model with $\nu$ species of Majorana fermions coupled to a static $\mathbb{Z}_2$ gauge field. Here each species of Majorana fermions gives rise to an energy band that carries a Chern number $\mathcal{C} = 1$, yielding a total Chern number $\mathcal{C} = \nu$. It has been shown that the ground states are three (four)-fold topologically degenerate on a torus, when $\nu$ is an odd (even) number. These exactly solvable models can be achieved by quantum simulations.



# 1 Introduction

Topological order [1–3] is a novel organizing principle for gapped quantum matters that is beyond the classic Landau-Ginzburg-Wilson paradigm. Instead of the spontaneous symmetry breaking in the classic paradigm, the long-range quantum entanglement plays an essential role in topological orders [4]. In a pioneer work [5], Kitaev proposed a systematic method, dubbed "sixteen-fold way", to characterize and classify topological orders in two dimensional (2D) quantum systems that consist of weakly interacting fermions. The basic idea is that the topological properties of a 2D gapped state can be uniquely characterized by the topological properties of its fractional quasi-particle excitations, "anyons", in bulk: namely, the distinct classes of anyons, and the statistics and the fusion rule among them. These bulk topological properties of anyons also encode information about possible chiral gapless edge states inherently.

The simplest example for topological order is a $\mathbb{Z}_2$ gauge theory [3], on which a gas of free (Majorana) fermions is coupled to a static $\mathbb{Z}_2$ gauge field. It was suggested by Kitaev that the crucial bulk parameter for the anyon statistic is the topological spin of a vortex, $\theta_\sigma = e^{i\pi\nu/8}$, where $\nu$ is the total Chern number of Majorana fermions. So that the topological properties depend only on $\nu$ mod 16, rather than the Chern number $\nu$ itself. The topological orders of Kitaev's sixteen-fold way are closely related to a wide range of topological phases of matter [6], including fractional quantum Hall insulators [7], topological superconductors, and quantum spin liquids [8–12]. In particular, Kitaev proposed an exactly solvable spin-1/2 model defined on a honeycomb lattice that can harbor $\nu = 0$ and $\nu = \pm 1$ topologically ordered states [5].

Besides the exact solution via the elegant four Majorana fermion decomposition method, which was proposed by Kitaev himself, the honeycomb spin-1/2 model can be exactly solved with the help of Jordan-Wigner transformation as well [13–16]. The Jordan-Winger transformation enables a fermionization of the spin model without redundant degrees of freedom (that are unavoidable in the four Majorana decomposition and can be removed by imposing a Gutzwiller projection), and allows us to map the original spin-1/2 model to a $p$-wave-spinless BCS pairing model. Thus, the weak pairing gives rise to the $\nu = \pm 1$ non-Abelian phase, while the strong pairing leads to the $\nu = 0$ Abelian phase [10].

Moreover, the Jordan-Wigner transformation also provides a topological characterization of quantum phases and quantum phase transition by itself: As long as a one-dimensional (1D) path has been properly chosen to "lace" all the sites on the 2D honeycomb lattice, which is the prerequisite for the Jordan-Winger transformation, a nonlocal string order parameter can be defined in one of the two phases ($\nu = 0$ and $\nu = \pm 1$) [14, 16]. These string order parameters become local order parameters after some singular transformation. In appropriate dual representations in the two phases, a description of the phase transition in terms of Landau's theory of continuous phase transitions becomes applicable [14].

Due to the significance of the exactly solvable honeycomb model, great theoretical efforts have been devoted to searching for its generalizations. The generalizations to other 2D and 3D lattice models can be found in Ref. [17–21] and Ref. [22–29], respectively. There are also some generalized models with multiple-spin interactions [30, 31]. The generalizations to higher spin models have been achieved in a $\Gamma$ matrix representation [32–35]. Recently, a class

of generalized Kitaev spin-1/2 models have been constructed in arbitrary dimensions, which can be solved exactly with the aid of the Jordan-Wigner transformation [31].

One of the most important issues on 2D topological orders is to find exactly solvable models for $|\nu| \geq 2$ classes. Indeed, some examples have been demonstrated in Ref. [36–38]. Moreover, a full construction of explicit lattice models for all $\nu$ mod 16 has been achieved in Ref. [39] very recently, on which a series of $\Gamma$ matrix models [17,33] have been proposed on honeycomb (square) lattice to realize odd (even) Chern number $\nu = 2q - 1$ ($\nu = 2q - 2$). However, a systematic construction for exactly solvable spin-1/2 models for all $\nu$ mod 16 is still in demand.

In this paper, we construct a family of quantum spin-1/2 models to realize Kitaev's sixteen-fold way for 2D topological orders that can be solved exactly via the Jordan-Wigner transformation. We have proposed two kinds of models on a 2D lattice which consists of $2q$ sites in each unit cell in accordance with even and odd Chern number $\nu$. By defining the ordering of lattice sites and performing the Jordan-Wigner transformation, we are able to map these spin-1/2 models to a $\mathbb{Z}_2$ gauge theory that consists of $\nu$ species of free Majorana fermions and a static $\mathbb{Z}_2$ gauge field. Since each species of Majorana fermions contributes a Chern number $\mathcal{C} = 1$, the whole system has a total Chern number $\mathcal{C} = \nu$.

The rest of this paper is organized as follows. We begin with revisiting the Kitaev honeycomb model and the Jordan-Wigner transformation in Section 2, which is the major technique used in the present work. Then a family of exactly solvable quantum spin-1/2 models have been constructed for odd and even Chern numbers in Section 3. In Section 4, we solve these spin model on a torus and study the ground state degeneracy that characterizes the topological orders. Section 5 is devoted to conclusions and discussions, where the equivalence relationship between our $\nu = 2$ model and the Yao-Zhang-Kivelson (YZK) model [32] has been discussed.

## 2 Solve the Kitaev honeycomb model via Jordan-Wigner transformation

The original Kitaev honeycomb model is defined by the following Hamiltonian,

$$H = -J_1 \sum_{\langle ij \rangle_x} \sigma_i^x \sigma_j^x - J_2 \sum_{\langle ij \rangle_y} \sigma_i^y \sigma_j^y - J_3 \sum_{\langle ij \rangle_z} \sigma_i^z \sigma_j^z, \tag{1}$$

where $i$ and $j$ label the sites on a honeycomb lattice as plotted in Fig. 1. $\sigma_i^\alpha$ ($\alpha = x, y, z$) is the Pauli matrix at site $i$, $\langle ij \rangle_\alpha$ denotes the nearest neighbor (NN) bond in the $\alpha$ direction, and $J_\lambda$ ($\lambda = 1, 2, 3$ for $\alpha = x, y, z$) is the corresponding coupling constant. In order to be consistent with the models constructed in Section 3, we use $J_\lambda$ rather than $J_\alpha$ here, which is different from the notation in Kitaev's original work.

As mentioned before, the Kitaev honeycomb model defined in Eq. (1) can be solved exactly by using the Jordan-Wigner transformation [13]. The exact solution can be done on different geometries, with either open boundary condition [14–16] or periodic boundary condition [18, 40] (PBC). To do this, we introduce the brick-wall representation of the honeycomb lattice [14, 15], and label each lattice site by the unit cell vector $\vec{r} = l_1 \vec{n}_1 + l_2 \vec{n}_2$ ($l_1, l_2 \in \mathbb{N}_+$) and the sublattice index $\beta = A, B$, where $\vec{n}_1$ and $\vec{n}_2$ are the primitive vectors for the brick-wall lattice [as shown in Fig. 1 (c)]. Notice that the primitive vectors that we choose here are slightly different from those in previous works. Then the Hamiltonian in Eq. (1) can be written as

$$H = -\sum_{\vec{r}} \left( J_1 \sigma_{\vec{r},A}^x \sigma_{\vec{r},B}^x + J_2 \sigma_{\vec{r},A}^y \sigma_{\vec{r}-\vec{n}_1,B}^y + J_3 \sigma_{\vec{r},A}^z \sigma_{\vec{r}-\vec{n}_1-\vec{n}_2,B}^z \right). \tag{2}$$

In order to perform the Jordan-Wigner transformation, we define a 1D path through all the lattice sites as follows: for two sites $l$ and $m$, (1) if $l_2 < m_2$, then $l < m$; (2) if $l_2 = m_2$

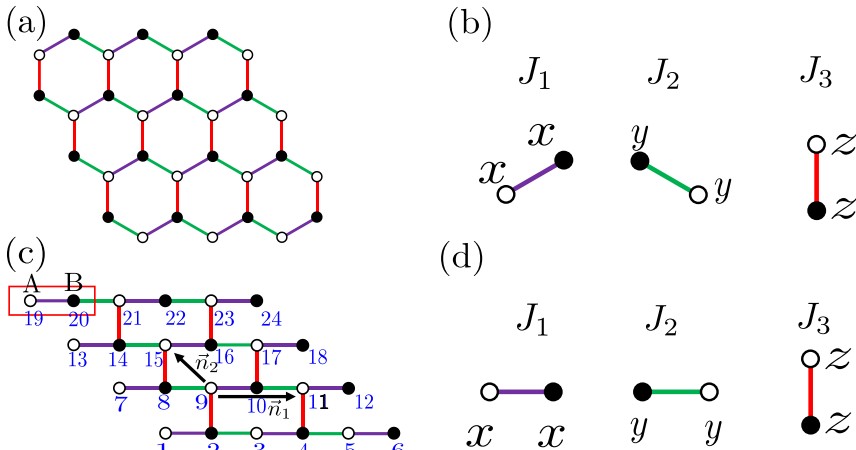

Figure 1: (a) A honeycomb lattice on which the Hamiltonian in Eq. (1) is defined. (b) The terms in Hamiltonian (1). (c) Brick-wall representation of the honeycomb lattice. A site is labeled by the unit cell vector $\vec{r} = l_1\vec{n}_1 + l_2\vec{n}_2$ ($l_1, l_2 \in \mathbb{N}_+$) and the sublattice index $\beta = A, B$, where $\vec{n}_1$ and $\vec{n}_2$ are the primitive vectors. White and black circles represent sublattices A and B, respectively. The ordering of sites is indicated by numbers that defines a 1D path for the Jordan-Wigner transformation. (d) The terms in Hamiltonian (2).

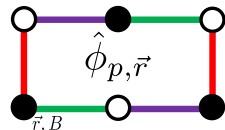

Figure 2: A plaquette where the flux operator $\hat{\phi}_{p,\vec{r}}$ in Eq. (5) is defined.

and $l_1 < m_1$, then $l < m$; (3) if $l_2 = m_2$, $l_1 = m_1$, and $l \in A$, $m \in B$, then $l < m$. By this definition of site ordering, we are able to solve the Hamiltonian defined in Eq. (2) with the help of the Jordan-Wigner transformation, which is given by

$$\sigma_m^+ = f_m^\dagger e^{i\pi \sum_{l<m} \hat{n}_l}, \tag{3a}$$

$$\sigma_m^z = 2\hat{n}_m - 1, \tag{3b}$$

where $\sigma_m^+ = \frac{1}{2}\left(\sigma_m^x + i\sigma_m^y\right)$ is the spin raising operator, $f_m^\dagger$ is the creation operator for the spinless fermion at site $m$, and $\hat{n}_m = f_m^\dagger f_m$ is the fermion occupation number operator. We further decompose each complex fermion $f_m$ into two Majorana fermions $\eta_m$ and $\gamma_m$ as follows: (1) for $m \in A$, $\eta_m = f_m^\dagger + f_m$ and $\gamma_m = i(f_m^\dagger - f_m)$; (2) for $m \in B$, $\eta_m = i(f_m^\dagger - f_m)$ and $\gamma_m = f_m^\dagger + f_m$. After the Jordan-Wigner transformation, the Hamiltonian (2) takes the form of

$$H = i \sum_{\vec{r}} \left( J_1 \gamma_{\vec{r},A} \gamma_{\vec{r},B} + J_2 \gamma_{\vec{r},A} \gamma_{\vec{r}-\vec{n}_1,B} + J_3 \hat{D}_{\vec{r}-\vec{n}_1-\vec{n}_2} \gamma_{\vec{r},A} \gamma_{\vec{r}-\vec{n}_1-\vec{n}_2,B} \right), \tag{4}$$

where $\hat{D}_{\vec{r}} = i\eta_{\vec{r},B}\eta_{\vec{r}+\vec{n}_1+\vec{n}_2,A}$. It is easy to verify that $\hat{D}_{\vec{r}}$ commute with each other and with the Hamiltonian (2), and $\hat{D}_{\vec{r}}^2 = 1$. So we can replace the operator $\hat{D}_{\vec{r}}$ by its eigenvalues $D_{\vec{r}} = \pm 1$, which can be viewed as a static $\mathbb{Z}_2$ gauge field. To characterize the $\mathbb{Z}_2$ gauge field, an alternative and gauge-invariant way is to define a flux operator $\hat{\phi}_{p,\vec{r}}$ on each plaquette, which is a specific product of 6 spin operators [as shown in Fig. 2],

$$\hat{\phi}_{p,\vec{r}} = \sigma_{\vec{r},B}^x \sigma_{\vec{r}+\vec{n}_1+\vec{n}_2,A}^y \sigma_{\vec{r}+\vec{n}_1+\vec{n}_2,B}^z \sigma_{\vec{r}+2\vec{n}_1+\vec{n}_2,A}^x \sigma_{\vec{r}+\vec{n}_1,B}^y \sigma_{\vec{r}+\vec{n}_1,A}^z$$

$$= \hat{D}_{\vec{r}} \hat{D}_{\vec{r}+\vec{n}_1}. \tag{5}$$

It is easy to see that $\hat{\phi}_{p,\vec{r}}^2 = 1$. So that the Hilbert space can be divided into subspaces in accordance with the sets of eigenvalues $\{\phi_{p,\vec{r}} = \pm 1\}$.

According to Lieb's theorem [41], the ground state of the Hamiltonian (2) is in the sector of Hilbert space on which $\hat{\phi}_{p,\vec{r}} = 1$ everywhere, i.e., the zero-flux sector. Thus we set all $D_{\vec{r}} = 1$ to solve Eq. (4) in the ground state sector and obtain the following energy dispersion,

$$\varepsilon\left(\vec{k}\right) = \pm 2 \left| J_1 + J_2 e^{-i\vec{k}\cdot\vec{n}_1} + J_3 e^{-i\vec{k}\cdot(\vec{n}_1+\vec{n}_2)} \right|. \tag{6}$$

As pointed out by Kitaev [5], the energy dispersion in Eq. (6) will be gapped if one of the three $|J_\lambda|$ is greater than the sum of the remaining two and will be gapless otherwise. The gapped phase yields a Chern number $\nu = 0$. In the gapless phase, a time-reversal-symmetry (TRS) breaking perturbation will open the gap at the Dirac cones of Eq. (6), and give rise to a Chern number $\nu = \pm 1$.

## 3 Exactly solvable spin-1/2 models for arbitrary Chern number

In the spirit of the separation of the degrees of freedom as discussed in the previous section, we generalize Kitaev honeycomb model to obtain exactly solvable spin-1/2 model for arbitrary Chern number $\nu$. First, we notice that there are two types of Majorana fermions in the Jordan-Wigner transformed Kitaev honeycomb model, i.e., "gauge Majorana fermions" and "itinerant Majorana fermions" [$\eta$- and $\gamma$- Majorana fermions in Eq. (4)]. Gauge Majorana fermions are localized on the vertical bonds of the brick-wall lattice [see $z$-$z$ bonds on Fig. 1] and give rise to the static $\mathbb{Z}_2$ gauge field $\hat{D}_{\vec{r}}$, while the itinerant Majorana fermions are non-interacting and coupled to $\hat{D}_{\vec{r}}$. Second, the honeycomb or brick-wall lattice can be divided into two sublattices, such that each unit cell consists of two (or an even number of) Majorana fermions on each sublattice (A or B), namely, one is $\eta$ and the other is $\gamma$.

It is sufficient to consider non-negative $\nu \geq 0$, since the topological order of Chern number $\mathcal{C} = -\nu$ is the Kramers counterpart of the $\mathcal{C} = \nu$ one. To realize the topological order with $\mathcal{C} = \nu \geq 1$, it is natural to make $\nu$ copies of itinerant Majorana fermions that are all coupled to a single static $\mathbb{Z}_2$ gauge field on a brick-wall-type lattice. Thus, we need $\nu$ pairs of itinerant Majorana fermions and at least one pair of gauge Majorana fermions per unit cell. When $\nu = 2q - 1$ ($q = 1, 2, \cdots$) is an odd number, only one pair of gauge Majorana fermions are required, and each unit cell consists of $2q = \nu + 1$ physical spins; while for an even Chern number $\nu = 2q - 2$ ($q = 2, 3, \cdots$), we need four gauge Majorana fermions, and there are $2q = \nu + 2$ physical spins per unit cell.

The above idea can be illustrated by the Kitaev honeycmb model itself: There are 4 Majorana fermions ($\gamma_A$, $\gamma_B$, $\eta_A$ and $\eta_B$) per unit cell in Kitaev honeycomb model. The Jordan-Wigner transformed Hamiltonian (4) contains a pair of itinerant Majorana fermions $\gamma_A$ and $\gamma_B$ that are coupled to a static $\mathbb{Z}_2$ gauge field described by $\hat{D}_{\vec{r}} = i\eta_{\vec{r},B}\eta_{\vec{r}+\vec{n}_1+\vec{n}_2,A}$. So that Kitaev honeycomb model is able to host topologically ordered ground states in the vortex free sector up to $|\nu| = 1$.[1]

Indeed, a "$\nu$ copies of itinerant Majorana fermions" construction has been proposed for higher Chern number $\nu \geq 2$ by Chulliparambil *et al.* in Ref. [39], which is a generalization of Kitaev's four-Majorana construction. They defined odd number $\nu = 2q - 1$ models on a honeycomb lattice and even number $\nu = 2q - 2$ models on a square lattice, where both square and honeycomb lattices are bipartite and have two sites per unit cell. To be specific, they

---

[1]Here a small time-reversal symmetry breaking perturbation can be added to open an energy gap. It is also worth mentioning that for large time-reversal symmetry breaking term, the ground-state is in the vortex full sector has a Chern number $\nu = 2$ [42].

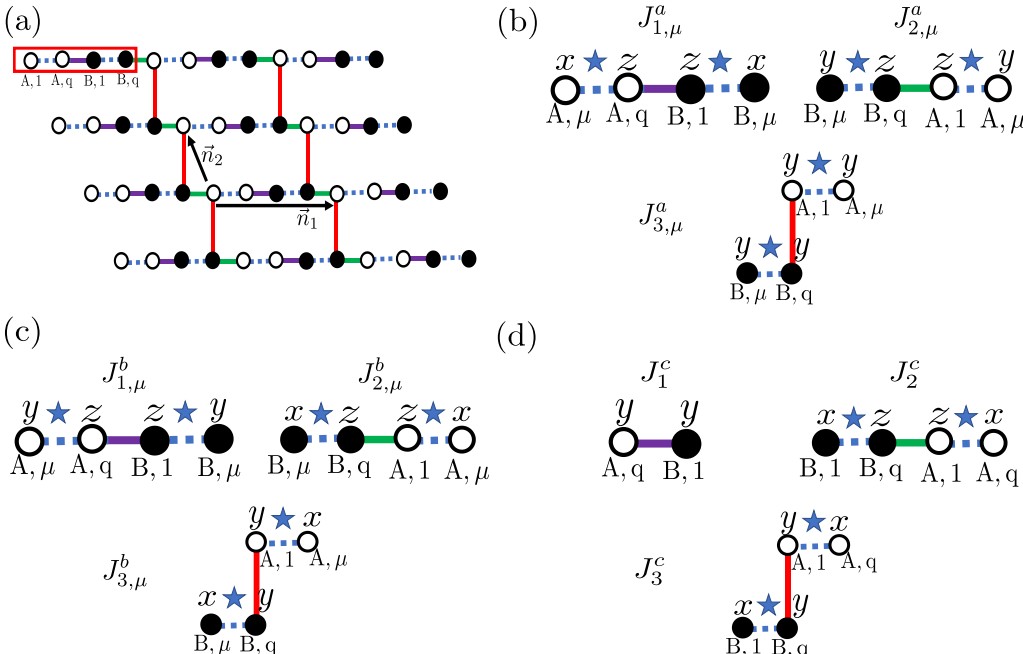

Figure 3: (a) A 2D lattice where the Hamiltonian (7) is defined. Each unit cell consists of $2q$ sites and dashed lines represent the abbreviated $q-2$ sites between the sites $\mu = 1$ and $\mu = q$. Each unit cell is labeled by $\vec{r} = l_1\vec{n}_1 + l_2\vec{n}_2$ ($l_1, l_2 \in \mathbb{N}_+$), where $\vec{n}_1$ and $\vec{n}_2$ are the primitive vectors for the lattice. The sites in a unit cell are divided into two sublattices, A and B. White circles and black circles represent sites in sublattice A and B, respectively. In each set, the index $\mu$ ($\mu = 1, 2, \ldots, q$) is used to distinguish different sites. (b), (c) and (d) denotes the terms in $H_a^{\text{odd}}$, $H_b^{\text{odd}}$ and $H_c^{\text{odd}}$, respectively. Blue stars represent the product of $\sigma^z$'s on corresponding abbreviated sites.

defined models by $\Gamma$ matrices [17, 33] that are $2^q$-dimensional representation of the Clifford algebra, say, a set of Hermitian matrices $\Gamma^\alpha$ ($\alpha = 1, \cdots, 2q + 1$) satisfying $\{\Gamma^\alpha, \Gamma^\beta\} = 2\delta_{\alpha\beta}$, together with their commutators $\Gamma^{\alpha\beta} = \frac{i}{2}[\Gamma^\alpha, \Gamma^\beta]$. For the purpose of solving these $\Gamma$-matrix models, $2q + 2$ Majorana fermions have been introduced to represent the $2^q$-dimensional Cliffordα agebra, i.e., $\Gamma_j^\alpha = ib_j^\alpha c_j$ and $\Gamma^{\alpha\beta} = ib_j^\alpha b_j^\beta$ ($\alpha, \beta = 1, \cdots, 2q + 1$). Note that the $2q + 2$ Majorana fermions double the local Hilbert space, so the local constraint $ic_j \prod_{\alpha=1}^{2q+1} b_j^\alpha = 1$ has to be imposed to restore the $2^q$-dimensional physical Hilbert space. The coordinate number of a honeycomb (square) lattice is $z = 3(4)$. So that $z$ species of Majorana fermions are localized to form a static $\mathbb{Z}_2$ gauge field, and the remaining $2q + 2 - z$ species Majorana fermions are mobile and give rise to a Chern number $\mathcal{C}$ up to $\nu = 2q + 2 - z$, namely, $\nu = 2q - 1$ on the honeycomb lattice and $\nu = 2q - 2$ on the square lattice.

In the rest part of this section, we shall construct spin-1/2 models for odd and even Chern numbers, say, $\nu = 2q-1$ and $\nu = 2q-2$ ($q = 2, 3, \cdots$), respectively. These spin-1/2 models can be exactly solved by the Jordan-Wigner transformation without redundant degrees of freedom.

## 3.1 Odd Chern number $\nu = 2q - 1$

To realize $\nu = 2q - 1$ ($q \geq 2$) topological orders, we shall construct spin-1/2 models on brick-wall lattices consisting of $2q$ sites per unit cell, such that the Jordan-Wigner will map these $2q$ spins to $4q$ Majorana fermions. Divide all the lattice sites into two sets ($A$ and $B$) and choose one species of Majorana fermions ($\eta$-Majorana fermions) on each sublattice ($A$ or $B$) to form a static $\mathbb{Z}_2$ gauge field, we are able to utilize the remaining $4q-2$ itinerant Majorana fermions to

construct $2q - 1$ copies of $\nu = 1$ Majorana fermion band whose Hamiltonian takes the pairing form of Eq. (4). Note that the $2q - 1$ copies of itinerant Majorana fermions are couple to the same static $\mathbb{Z}_2$ gauge field that arises from gauge Majorana fermions. In order to lift possible local degeneracy, as pointed out in Ref. [31], the coupling between two Majorana fermions on different sites can be achieved by introducing a spin-1/2 string operators connecting the two sites.

We define our models on a $L_1 \times L_2 \times 2q$ brick-wall lattice as plotted in Fig. 3 (a), which can be divided into two sublattices, $A$ and $B$. Introducing a basis index $\mu$, one can label a lattice site as $(\vec{r}, \beta, \mu)$, where $\vec{r}$ is the Bravais lattice vector, $\beta = A, B$, and $\mu = 1, 2, \cdots, q$. The Hamiltonian $H^{\text{odd}}$ takes a form of

$$H^{\text{odd}} = H_a^{\text{odd}} + H_b^{\text{odd}} + H_c^{\text{odd}}. \tag{7a}$$

Here the three parts in $H^{\text{odd}}$: $H_a^{\text{odd}}$, $H_b^{\text{odd}}$ and $H_c^{\text{odd}}$ can be written as follows,

$$
\begin{aligned}
H_a^{\text{odd}} = (-1)^q \sum_{\vec{r}} \sum_{\mu} \Bigg( & J_{1,\mu}^a \sigma_{\vec{r},A,\mu}^x \left( \prod_{\mu+1 \leqslant \rho \leqslant q} \sigma_{\vec{r},A,\rho}^z \right) \left( \prod_{1 \leqslant \rho \leqslant \mu-1} \sigma_{\vec{r},B,\rho}^z \right) \sigma_{\vec{r},B,\mu}^x \\
& + J_{2,\mu}^a \sigma_{\vec{r}-\vec{n}_1,B,\mu}^y \left( \prod_{\mu+1 \leqslant \rho \leqslant q} \sigma_{\vec{r}-\vec{n}_1,B,\rho}^z \right) \left( \prod_{1 \leqslant \rho \leqslant \mu-1} \sigma_{\vec{r},A,\rho}^z \right) \sigma_{\vec{r},A,\mu}^y \\
& + J_{3,\mu}^a \sigma_{\vec{r}-\vec{n}_1-\vec{n}_2,B,\mu}^y \left( \prod_{\mu+1 \leqslant \rho \leqslant q-1} \sigma_{\vec{r}-\vec{n}_1-\vec{n}_2,B,\rho}^z \right) \sigma_{\vec{r}-\vec{n}_1-\vec{n}_2,B,q}^y \sigma_{\vec{r},A,1}^y \left( \prod_{2 \leqslant \rho \leqslant \mu-1} \sigma_{\vec{r},A,\rho}^z \right) \sigma_{\vec{r},A,\mu}^y \Bigg),
\end{aligned}
\tag{7b}
$$

$$
\begin{aligned}
H_b^{\text{odd}} = (-1)^{q+1} \sum_{\vec{r}} \sum_{\mu}' \Bigg( & J_{1,\mu}^b \sigma_{\vec{r},A,\mu}^y \left( \prod_{\mu+1 \leqslant \rho \leqslant q} \sigma_{\vec{r},A,\rho}^z \right) \left( \prod_{1 \leqslant \rho \leqslant \mu-1} \sigma_{\vec{r},B,\rho}^z \right) \sigma_{\vec{r},B,\mu}^y \\
& + J_{2,\mu}^b \sigma_{\vec{r}-\vec{n}_1,B,\mu}^x \left( \prod_{\mu+1 \leqslant \rho \leqslant q} \sigma_{\vec{r}-\vec{n}_1,B,\rho}^z \right) \left( \prod_{1 \leqslant \rho \leqslant \mu-1} \sigma_{\vec{r},A,\rho}^z \right) \sigma_{\vec{r},A,\mu}^x \\
& + J_{3,\mu}^b \sigma_{\vec{r}-\vec{n}_1-\vec{n}_2,B,\mu}^x \left( \prod_{\mu+1 \leqslant \rho \leqslant q-1} \sigma_{\vec{r}-\vec{n}_1-\vec{n}_2,B,\rho}^z \right) \sigma_{\vec{r}-\vec{n}_1-\vec{n}_2,B,q}^y \sigma_{\vec{r},A,1}^y \left( \prod_{2 \leqslant \rho \leqslant \mu-1} \sigma_{\vec{r},A,\rho}^z \right) \sigma_{\vec{r},A,\mu}^x \Bigg),
\end{aligned}
\tag{7c}
$$

$$
\begin{aligned}
H_c^{\text{odd}} = \sum_{\vec{r}} \Bigg( & J_1^c \sigma_{\vec{r},A,q}^y \sigma_{\vec{r},B,1}^y + J_2^c \sigma_{\vec{r}-\vec{n}_1,B,1}^x \left( \prod_{2 \leqslant \rho \leqslant q} \sigma_{\vec{r}-\vec{n}_1,B,\rho}^z \right) \left( \prod_{1 \leqslant \rho \leqslant q-1} \sigma_{\vec{r},A,\rho}^z \right) \sigma_{\vec{r},A,q}^x \\
& + J_3^c \sigma_{\vec{r}-\vec{n}_1-\vec{n}_2,B,1}^x \left( \prod_{2 \leqslant \rho \leqslant q-1} \sigma_{\vec{r}-\vec{n}_1-\vec{n}_2,B,\rho}^z \right) \sigma_{\vec{r}-\vec{n}_1-\vec{n}_2,B,q}^y \sigma_{\vec{r},A,1}^y \left( \prod_{2 \leqslant \rho \leqslant q-1} \sigma_{\vec{r},A,\rho}^z \right) \sigma_{\vec{r},A,q}^x \Bigg),
\end{aligned}
\tag{7d}
$$

where $\sigma_{\vec{r}_l,\beta_l,\mu_l}^\alpha$ $(\alpha = x, y, z)$ is the Pauli matrix at site $l$. The coupling constant $J_{\lambda,\mu}^{a(b)}$ $(\lambda = 1, 2, 3)$ in $H_{a(b)}^{\text{odd}}$ is associated with a string of operators that connects two sites sharing the same $\mu$ index. The $J_\lambda^c$ terms in $H_c^{\text{odd}}$ couple two sites with indices $(A, q)$ and $(B, 1)$ via a string of operators. The summations $\sum_\mu = \sum_{\mu=1}^q$ and $\sum_\mu' = \sum_{\mu=2}^{q-1}$, respectively. Two conventions have been adopted in Eq. (7): (i) $\left( \prod_{l \leqslant \rho \leqslant m} \sigma_{\vec{r},\beta,\rho}^z \right) \equiv 1$ if $m \leqslant l$ and (ii) $\sigma_{\vec{r},A,1}^y \left( \prod_{2 \leqslant \rho \leqslant \mu-1} \sigma_{\vec{r},A,\rho}^z \right) \sigma_{\vec{r},A,\mu}^y \equiv -\sigma_{\vec{r},A,1}^z$ if $\mu = 1$. $H_a^{\text{odd}}$ and $H_b^{\text{odd}}$ consist of $(q+1)$-spin interactions only and the terms in $H_c^{\text{odd}}$ are of different lengths.

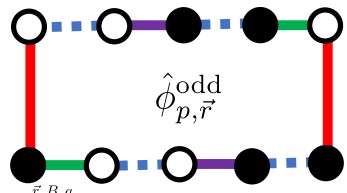

Figure 4: A plaquette where the flux operator $\hat{\phi}^{\text{odd}}_{p,\vec{r}}$ in Eq. (8) is defined.

The Hamiltonian (7) has flux operators as local integrals of motion, which is similar to those in Kitaev honeycomb model. The flux operator $\hat{\phi}^{\text{odd}}_{p,\vec{r}}$ is defined as follows [see Fig. 4],

$$
\begin{aligned}
\hat{\phi}^{\text{odd}}_{p,\vec{r}} =& \sigma^x_{\vec{r},B,q} \sigma^y_{\vec{r}+\vec{n}_1+\vec{n}_2,A,1} \left( \prod_{2 \leqslant \rho \leqslant q} \sigma^z_{\vec{r}+\vec{n}_1+\vec{n}_2,A,\rho} \right) \left( \prod_{1 \leqslant \rho \leqslant q} \sigma^z_{\vec{r}+\vec{n}_1+\vec{n}_2,B,\rho} \right) \\
& \times \sigma^x_{\vec{r}+2\vec{n}_1+\vec{n}_2,A,1} \sigma^y_{\vec{r}+\vec{n}_1,B,q} \left( \prod_{1 \leqslant \rho \leqslant q} \sigma^z_{\vec{r}+\vec{n}_1,A,\rho} \right) \left( \prod_{1 \leqslant \rho \leqslant q-1} \sigma^z_{\vec{r}+\vec{n}_1,B,\rho} \right).
\end{aligned}
\tag{8}
$$

Because all flux operators $\hat{\phi}^{\text{odd}}_{p,\vec{r}}$ commute with each other and with the Hamiltonian (7), and $\left( \hat{\phi}^{\text{odd}}_{p,\vec{r}} \right)^2 = 1$. The eigenvalues of $\hat{\phi}^{\text{odd}}_{p,\vec{r}}$ compose a set of good quantum numbers $\left\{ \phi^{\text{odd}}_{p,\vec{r}} \right\}$, where $\phi^{\text{odd}}_{p,\vec{r}} = \pm 1$ is the eigenvalue of the flux operator $\hat{\phi}^{\text{odd}}_{p,\vec{r}}$.

The Jordan-Wigner transformation will be exploited to solve the model defined in Eq. (7) in accordance to the sets of good quantum numbers $\left\{ \phi^{\text{odd}}_{p,\vec{r}} \right\}$. To implement it, we keep the three sort rules given in Section 2 unchanged and add a fourth sort rule associated with the basis index $\mu$: (4) if $l_1 = m_1$, $l_2 = m_2$, $\beta_l = \beta_m$ and $\mu_l < \mu_m$, then $l < m$. The way to decompose complex fermion $f_m$ depends on the sublattice index $\beta_m$: (1) for $\beta_m = A$, $\eta_m = f_m^\dagger + f_m$ and $\gamma_m = i(f_m^\dagger - f_m)$; (2) for $\beta_m = B$, $\eta_m = i(f_m^\dagger - f_m)$ and $\gamma_m = f_m^\dagger + f_m$. The Hamiltonians $H^{\text{odd}}_{a,b,c}$ are mapped to be

$$
H^{\text{odd}}_a = i \sum_{\vec{r}} \sum_\mu \left( J^a_{1,\mu} \gamma_{\vec{r},A,\mu} \gamma_{\vec{r},B,\mu} + J^a_{2,\mu} \gamma_{\vec{r},A,\mu} \gamma_{\vec{r}-\vec{n}_1,B,\mu} + J^a_{3,\mu} \hat{D}_{\vec{r}-\vec{n}_1-\vec{n}_2} \gamma_{\vec{r},A,\mu} \gamma_{\vec{r}-\vec{n}_1-\vec{n}_2,B,\mu} \right),
\tag{9a}
$$

$$
H^{\text{odd}}_b = i \sum_{\vec{r}} \sum_\mu{}' \left( J^b_{1,\mu} \eta_{\vec{r},A,\mu} \eta_{\vec{r},B,\mu} + J^b_{2,\mu} \eta_{\vec{r},A,\mu} \eta_{\vec{r}-\vec{n}_1,B,\mu} + J^b_{3,\mu} \hat{D}_{\vec{r}-\vec{n}_1-\vec{n}_2} \eta_{\vec{r},A,\mu} \eta_{\vec{r}-\vec{n}_1-\vec{n}_2,B,\mu} \right),
\tag{9b}
$$

$$
H^{\text{odd}}_c = i \sum_{\vec{r}} \left( J^c_1 \eta_{\vec{r},A,q} \eta_{\vec{r},B,1} + J^c_2 \eta_{\vec{r},A,q} \eta_{\vec{r}-\vec{n}_1,B,1} + J^c_3 \hat{D}_{\vec{r}-\vec{n}_1-\vec{n}_2} \eta_{\vec{r},A,q} \eta_{\vec{r}-\vec{n}_1-\vec{n}_2,B,1} \right),
\tag{9c}
$$

where $\hat{D}_{\vec{r}} = i \eta_{\vec{r},B,q} \eta_{\vec{r}+\vec{n}_1+\vec{n}_2,A,1}$ commute with each other and with the Hamiltonian $H^{\text{odd}}$, and $\hat{D}^2_{\vec{r}} = 1$. It can be seen from Eqs. (9) that the two Majorana fermions $\eta_{\vec{r},A,1}$ and $\eta_{\vec{r},B,q}$ constitute a static $\mathbb{Z}_2$ gauge field. The corresponding $\mathbb{Z}_2$ flux is given by $\hat{\phi}^{\text{odd}}_{p,\vec{r}} = \hat{D}_{\vec{r}} \hat{D}_{\vec{r}+\vec{n}_1}$.

For each sublattice $A$ or $B$, there exist $q$ species of itinerant Majorana fermions [$\gamma_{A,\mu}$ and $\gamma_{B,\mu}$ ($\mu = 1, 2, \ldots, q$)] in $H^{\text{odd}}_a$, $q - 2$ species of itinerant Majorana fermions [$\eta_{A,\mu}$ and $\eta_{B,\mu}$ ($\mu = 2, 3, \ldots, q-1$)] in $H^{\text{odd}}_b$, and one species of itinerant Majorana fermions [$\eta_{A,q}$ or $\eta_{B,1}$] in $H^{\text{odd}}_c$, respectively. All the $2q - 1$ species of itinerant Majorana fermions are coupled to the same $\mathbb{Z}_2$ gauge field. Moreover, the pairing of each species of itinerant Majorana fermions is of the same form as that in Eq. (4). Therefore, the ground state of $H^{\text{odd}}$ must be a zero flux state on which all the $\mathbb{Z}_2$ fluxes $\phi^{\text{odd}}_{p,\vec{r}} = 1$. The energy dispersion in the ground state sector reads

$$
\varepsilon^{\text{odd}}_{a(b),\mu}(\vec{k}) = \pm 2 \left| J^{a(b)}_{1,\mu} + J^{a(b)}_{2,\mu} e^{-i\vec{k}\cdot\vec{n}_1} + J^{a(b)}_{3,\mu} e^{-i\vec{k}\cdot(\vec{n}_1+\vec{n}_2)} \right|,
\tag{10a}
$$

$$
\varepsilon^{\text{odd}}_c(\vec{k}) = \pm 2 \left| J^c_1 + J^c_2 e^{-i\vec{k}\cdot\vec{n}_1} + J^c_3 e^{-i\vec{k}\cdot(\vec{n}_1+\vec{n}_2)} \right|.
\tag{10b}
$$

To obtain a topologically ordered state with an odd Chern number $\nu = 2q - 1$, we tune the coupling constants in the Hamiltonian $H^{\text{odd}}$ to make each filled bands in Eqs. (10) to have two Dirac cones as the gapless phase in the original Kitaev model [5]. Then a TRS breaking perturbation $H'^{\text{odd}}$ will be introduced to gap out all the Dirac cones. As long as the $2q - 1$ species of itinerant Majorana fermions remain decoupled in the presence of $H'^{\text{odd}}$, the total Chern number can be obtained by the sum over all the $2q - 1$ filled bands, resulting in $\nu = 2q - 1$. To simplify the discussion, we set all the coupling constants in $H^{\text{odd}}$ to be a single value $J$, and define the perturbation

$$H'^{\text{odd}} = H_a' + H_b' + H_c', \tag{11a}$$

that is composed of three parts as follows:

$$
\begin{aligned}
H_a' = \kappa \sum_{\vec{r}} \sum_{\mu} \Bigg( & \sigma_{\vec{r}-\vec{n}_1, A, \mu}^x \left( \prod_{\mu+1 \leqslant \rho \leqslant q} \sigma_{\vec{r}-\vec{n}_1, A, \rho}^z \right) \left( \prod_{1 \leqslant \rho \leqslant q} \sigma_{\vec{r}-\vec{n}_1, B, \rho}^z \right) \left( \prod_{1 \leqslant \rho \leqslant \mu-1} \sigma_{\vec{r}, A, \rho}^z \right) \sigma_{\vec{r}, A, \mu}^y \\
& + \sigma_{\vec{r}-\vec{n}_1, B, \mu}^y \left( \prod_{\mu+1 \leqslant \rho \leqslant q} \sigma_{\vec{r}-\vec{n}_1, B, \rho}^z \right) \left( \prod_{1 \leqslant \rho \leqslant q} \sigma_{\vec{r}, A, \rho}^z \right) \left( \prod_{1 \leqslant \rho \leqslant \mu-1} \sigma_{\vec{r}, B, \rho}^z \right) \sigma_{\vec{r}, B, \mu}^x \Bigg),
\end{aligned}
\tag{11b}
$$

$$
\begin{aligned}
H_b' = -\kappa \sum_{\vec{r}} {\sum_{\mu}}' \Bigg( & \sigma_{\vec{r}-\vec{n}_1, A, \mu}^y \left( \prod_{\mu+1 \leqslant \rho \leqslant q} \sigma_{\vec{r}-\vec{n}_1, A, \rho}^z \right) \left( \prod_{1 \leqslant \rho \leqslant q} \sigma_{\vec{r}-\vec{n}_1, B, \rho}^z \right) \left( \prod_{1 \leqslant \rho \leqslant \mu-1} \sigma_{\vec{r}, A, \rho}^z \right) \sigma_{\vec{r}, A, \mu}^x \\
& + \sigma_{\vec{r}-\vec{n}_1, B, \mu}^x \left( \prod_{\mu+1 \leqslant \rho \leqslant q} \sigma_{\vec{r}-\vec{n}_1, B, \rho}^z \right) \left( \prod_{1 \leqslant \rho \leqslant q} \sigma_{\vec{r}, A, \rho}^z \right) \left( \prod_{1 \leqslant \rho \leqslant \mu-1} \sigma_{\vec{r}, B, \rho}^z \right) \sigma_{\vec{r}, B, \mu}^y \Bigg),
\end{aligned}
\tag{11c}
$$

$$
\begin{aligned}
H_c' = -\kappa \sum_{\vec{r}} \Bigg( & \sigma_{\vec{r}-\vec{n}_1, A, q}^y \left( \prod_{1 \leqslant \rho \leqslant q} \sigma_{\vec{r}-\vec{n}_1, B, \rho}^z \right) \left( \prod_{1 \leqslant \rho \leqslant q-1} \sigma_{\vec{r}, A, \rho}^z \right) \sigma_{\vec{r}, A, q}^x \\
& + \sigma_{\vec{r}-\vec{n}_1, B, 1}^x \left( \prod_{2 \leqslant \rho \leqslant q} \sigma_{\vec{r}-\vec{n}_1, B, \rho}^z \right) \left( \prod_{1 \leqslant \rho \leqslant q} \sigma_{\vec{r}, A, \rho}^z \right) \sigma_{\vec{r}, B, 1}^y \Bigg).
\end{aligned}
\tag{11d}
$$

Note that all the terms in $H'^{\text{odd}}$ are of $(2q + 1)$-spin interactions. The Jordan-Wigner transformation will map $H'^{\text{odd}}$ to a quadratic form of Majorana fermions as follows,

$$
\begin{aligned}
H'^{\text{odd}} = i\kappa \sum_{\vec{r}} \Bigg( & \sum_{\mu} \left( \gamma_{\vec{r}, A, \mu} \gamma_{\vec{r}-\vec{n}_1, A, \mu} - \gamma_{\vec{r}, B, \mu} \gamma_{\vec{r}-\vec{n}_1, B, \mu} \right) \\
& + {\sum_{\mu}}' \left( \eta_{\vec{r}, A, \mu} \eta_{\vec{r}-\vec{n}_1, A, \mu} - \eta_{\vec{r}, B, \mu} \eta_{\vec{r}-\vec{n}_1, B, \mu} \right) + \left( \eta_{\vec{r}, A, q} \eta_{\vec{r}-\vec{n}_1, A, q} - \eta_{\vec{r}, B, 1} \eta_{\vec{r}-\vec{n}_1, B, 1} \right) \Bigg).
\end{aligned}
$$

Therefore the perturbed system $H^{\text{odd}} + H'^{\text{odd}}$ remains exactly solvable, and leads to gapped energy dispersions:

$$\varepsilon_{a(b), \mu}^{\text{odd}}(\vec{k}) = \varepsilon_c^{\text{odd}}(\vec{k}) = \pm 2 \sqrt{J^2 \left| 1 + e^{-i\vec{k} \cdot \vec{n}_1} + e^{-i\vec{k} \cdot (\vec{n}_1 + \vec{n}_2)} \right|^2 + \Delta^2(\vec{k})}, \tag{12}$$

where $\Delta(\vec{k}) = 2\kappa \sin(\vec{k} \cdot \vec{n}_1)$. Each filled band in Eq. (12) gives rise to a Chern number $\nu = \text{sgn}(\kappa)$, and the total Chern number is $\nu = (2q - 1)\text{sgn}(\kappa) = \pm(2q - 1)$.

It is remarkable that such a single-value choice of coupling constants leads to a model on which the SO$(2q - 1)$ symmetry is respected [39]. Nevertheless, we can adjust the couplings $J_{1(2,3), \mu}^{a(b)}$ and $J_{1(2,3)}^c$ in $H^{\text{odd}}$, such that some of the $2q - 1$ species of Majorana fermions remain

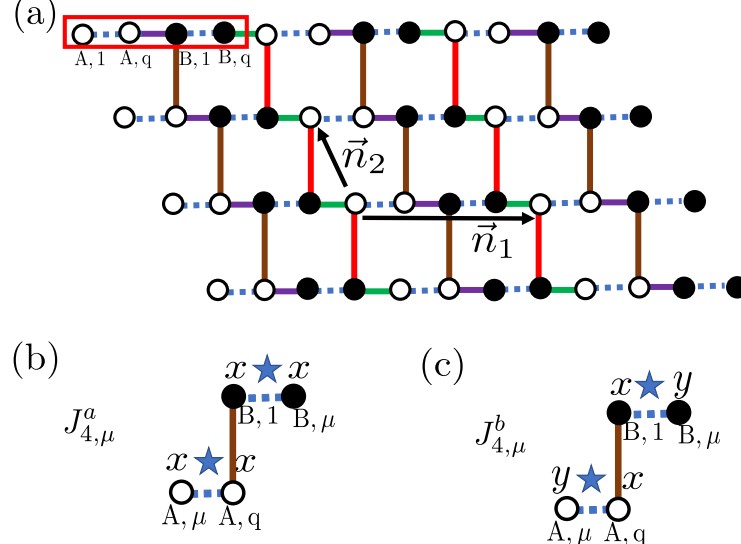

Figure 5: (a) A 2D lattice where the Hamiltonian (13) is defined. A bond in brown is added to connect the sites $\vec{r}$, $A$, $q$ and $\vec{r} + \vec{n}_2$, $B$, 1 in each plaquette of the former 2D lattice. (b) and (c) denotes the $J_{4,\mu}^a$ terms and $J_{4,\mu}^b$ terms in $H_a^{\text{even}}$ and $H_b^{\text{even}}$, respectively. Blue stars represent the product of $\sigma^z$'s on corresponding abbreviated sites.

gapless with Dirac cones, meanwhile other species are fully gapped by $H^{\text{odd}}$ itself, i.e. in the Abelian phase [5]. A small TRS breaking perturbation can open an energy gap at each Dirac cone, resulting in a non-Abelian state, on which each species of Majorana fermions gives rise to a Chern number $\nu = \pm 1$. Thus, we are able to realize $\nu = -2q+1, -2q+2, \cdots, 2q-2, 2q-1$ topological orders in different phases of a single exactly solvable model, while the $SO(2q-1)$ symmetry is no longer respected when $|\nu| < 2q-1$.

## 3.2 Even Chern number $\nu = 2q-2$

We proceed to construct spin-1/2 models that host $\nu = 2q-2$ topological orders and respect $SO(2q-2)$ symmetry. The $\nu = 2q-2$ model will be defined on a similar brick-wall lattice as the $\nu = 2q-1$ one, which consists of $2q$ sites per unit cell as well. The Major difference is that we need two rather than one gauge Majorana fermions per sublattice and per unit cell for $\nu = 2q-2$. Since gauge Majorana fermions are associated with vertical bonds, we need two vertical bonds per unit cell for $\nu = 2q-2$. The $\nu = 2q-2$ brick-wall lattice can be found in Fig. 5 (a), where additional vertical bonds connect sites $(\vec{r}, A, q)$ and $(\vec{r} + \vec{n}_2, B, 1)$. The model Hamiltonian

$$H^{\text{even}} = H_a^{\text{even}} + H_b^{\text{even}}, \tag{13a}$$

consists of two parts,

$$
\begin{aligned}
H_a^{\text{even}} = & H_a^{\text{odd}} + (-1)^{q+1} \sum_{\vec{r}} \sum_{\mu} J_{4,\mu}^a \sigma_{\vec{r},A,\mu}^x \left( \prod_{\mu+1 \leqslant \rho \leqslant q-1} \sigma_{\vec{r},A,\rho}^z \right) \\
& \times \sigma_{\vec{r},A,q}^x \sigma_{\vec{r}+\vec{n}_2,B,1}^x \left( \prod_{2 \leqslant \rho \leqslant \mu-1} \sigma_{\vec{r}+\vec{n}_2,B,\rho}^z \right) \sigma_{\vec{r}+\vec{n}_2,B,\mu}^x,
\end{aligned}
\tag{13b}
$$

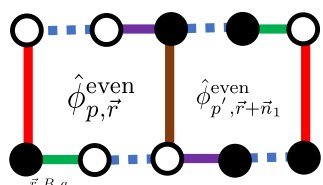

Figure 6: The plaquettes where the flux operators $\hat{\phi}^{\text{even}}_{p,\vec{r}}$ and $\hat{\phi}^{\text{even}}_{p',\vec{r}}$ in Eqs. (14) are defined.

$$
\begin{aligned}
H_b^{\text{even}} = &H_b^{\text{odd}} + (-1)^q \sum_{\vec{r}} {\sum_{\mu}}' J^b_{4,\mu} \sigma^y_{\vec{r},A,\mu} \left( \prod_{\mu+1 \leqslant \rho \leqslant q-1} \sigma^z_{\vec{r},A,\rho} \right) \\
&\times \sigma^x_{\vec{r},A,q} \sigma^x_{\vec{r}+\vec{n}_2,B,1} \left( \prod_{2 \leqslant \rho \leqslant \mu-1} \sigma^z_{\vec{r}+\vec{n}_2,B,\rho} \right) \sigma^y_{\vec{r}+\vec{n}_2,B,\mu} ,
\end{aligned}
\tag{13c}
$$

where extra terms with coupling constants $J^{a(b)}_{4,\mu}$ have been added to $H^{\text{odd}}_{a(b)}$, and if $\mu = 1$, $\sigma^x_{\vec{r}+\vec{n}_2,B,1} \left( \prod_{2 \leqslant \rho \leqslant \mu-1} \sigma^z_{\vec{r}+\vec{n}_2,B,\rho} \right) \sigma^x_{\vec{r}+\vec{n}_2,B,\mu} \equiv -\sigma^z_{\vec{r}+\vec{n}_2,B,1}$ . Note that each term in $H^{\text{even}}$ is of $q + 1$-spin interaction.

As shown in Fig. 6, there are two types of fluxes, $\hat{\phi}^{\text{even}}_{p,\vec{r}}$ and $\hat{\phi}^{\text{even}}_{p',\vec{r}}$, that are local integrals of motion of the Hamiltonian $H^{\text{even}}$. The operators $\hat{\phi}^{\text{even}}_{p,\vec{r}}$ and $\hat{\phi}^{\text{even}}_{p',\vec{r}}$ can be defined as follows,

$$
\hat{\phi}^{\text{even}}_{p,\vec{r}} = \sigma^x_{\vec{r},B,q} \sigma^y_{\vec{r}+\vec{n}_1+\vec{n}_2,A,1} \left( \prod_{2 \leqslant \rho \leqslant q} \sigma^z_{\vec{r}+\vec{n}_1+\vec{n}_2,A,\rho} \right) \sigma^y_{\vec{r}+\vec{n}_1+\vec{n}_2,B,1} \sigma^x_{\vec{r}+\vec{n}_1,A,q} \left( \prod_{1 \leqslant \rho \leqslant q-1} \sigma^z_{\vec{r}+\vec{n}_1,A,\rho} \right), \tag{14a}
$$

$$
\hat{\phi}^{\text{even}}_{p',\vec{r}} = \sigma^y_{\vec{r},A,q} \sigma^x_{\vec{r}+\vec{n}_2,B,1} \left( \prod_{2 \leqslant \rho \leqslant q} \sigma^z_{\vec{r}+\vec{n}_2,B,\rho} \right) \sigma^x_{\vec{r}+\vec{n}_1+\vec{n}_2,A,1} \sigma^y_{\vec{r},B,q} \left( \prod_{1 \leqslant \rho \leqslant q-1} \sigma^z_{\vec{r},B,\rho} \right). \tag{14b}
$$

All the flux operators $\hat{\phi}^{\text{even}}_{p,\vec{r}}$ and $\hat{\phi}^{\text{even}}_{p',\vec{r}}$ commute with each other and with $H^{\text{even}}$, whose eigenvalues $\phi^{\text{even}}_{p(p'),\vec{r}} = \pm 1$. So that the total Hilbert space for $H^{\text{even}}$ can be factorized into a direct product of sectors that are eigenspaces of $\hat{\phi}^{\text{even}}_{p,\vec{r}}$ and $\hat{\phi}^{\text{even}}_{p',\vec{r}}$.

Keeping the same ordering of sites as that for $H^{\text{odd}}$, we apply the same Jordan-Wigner transformation to $H^{\text{even}}$. Thus, the spin Hamiltonian $H^{\text{even}}$ in Eqs. (13) can be expressed in terms of Majorana fermions as follow,

$$
\begin{aligned}
H_a^{\text{even}} = i \sum_{\vec{r}} \sum_{\mu} &\Big( J^a_{1,\mu} \gamma_{\vec{r},A,\mu} \gamma_{\vec{r},B,\mu} + J^a_{2,\mu} \gamma_{\vec{r},A,\mu} \gamma_{\vec{r}-\vec{n}_1,B,\mu} \\
&+ J^a_{3,\mu} \hat{D}_{\vec{r}-\vec{n}_1-\vec{n}_2} \gamma_{\vec{r},A,\mu} \gamma_{\vec{r}-\vec{n}_1-\vec{n}_2,B,\mu} + J^a_{4,\mu} \hat{D}'_{\vec{r}} \gamma_{\vec{r},A,\mu} \gamma_{\vec{r}+\vec{n}_2,B,\mu} \Big),
\end{aligned}
\tag{15a}
$$

$$
\begin{aligned}
H_b^{\text{even}} = i \sum_{\vec{r}} {\sum_{\mu}}' &\Big( J^b_{1,\mu} \eta_{\vec{r},A,\mu} \eta_{\vec{r},B,\mu} + J^b_{2,\mu} \eta_{\vec{r},A,\mu} \eta_{\vec{r}-\vec{n}_1,B,\mu} \\
&+ J^b_{3,\mu} \hat{D}_{\vec{r}-\vec{n}_1-\vec{n}_2} \eta_{\vec{r},A,\mu} \eta_{\vec{r}-\vec{n}_1-\vec{n}_2,B,\mu} + J^b_{4,\mu} \hat{D}'_{\vec{r}} \eta_{\vec{r},A,\mu} \eta_{\vec{r}+\vec{n}_2,B,\mu} \Big),
\end{aligned}
\tag{15b}
$$

where $\hat{D}'_{\vec{r}} = i\eta_{\vec{r},A,q} \eta_{\vec{r}+\vec{n}_2,B,1}$. In parallel with the analyses for $\nu = 2q - 1$, bond operators $\hat{D}_{\vec{r}}$ and $\hat{D}'_{\vec{r}}$ commute with each other and with $H^{\text{even}}$, whose eigenvalues are $D_{\vec{r}}, D'_{\vec{r}} = \pm 1$. Thus $\hat{D}_{\vec{r}}$ and $\hat{D}'_{\vec{r}}$ constitute as a static $\mathbb{Z}_2$ gauge field together. The corresponding $\mathbb{Z}_2$ fluxes can be characterized by the flux operators defined in Eqs. (14) that read $\hat{\phi}^{\text{even}}_{p,\vec{r}} = \hat{D}_{\vec{r}} \hat{D}'_{\vec{r}+\vec{n}_1}$ and $\hat{\phi}^{\text{even}}_{p',\vec{r}} = \hat{D}_{\vec{r}} \hat{D}'_{\vec{r}}$. It is easy to verify that $\hat{\phi}^{\text{odd}}_{p,\vec{r}} = \hat{\phi}^{\text{even}}_{p,\vec{r}} \hat{\phi}^{\text{even}}_{p',\vec{r}+\vec{n}_1}$, which is implied in Fig. 6.

It can be seen from Eqs. (15) that $H_a^{\text{even}}$ describes $q$ species of itinerant Majorana fermions $\gamma_{A,\mu}$ and $\gamma_{B,\mu}$ ($\mu = 1, 2, \ldots, q$) and $H_b^{\text{even}}$ describes $q-2$ species of itinerant Majorana fermions

$\eta_{A,\mu}$ and $\eta_{B,\mu}$ ($\mu = 2, 3, \ldots, q-1$), respectively. All the $2q-2$ species of itinerant Majorana fermions are coupled to the same $\mathbb{Z}_2$ gauge field. Besides, the Hamiltonian for each species of the itinerant Majorana fermions is equivalent to a Hamiltonian defined on a square lattice whose unit cell consists of two sites $(\vec{r}, A, \mu)$ and $(\vec{r}, B, \mu)$. Thus the Lieb's theorem [41] for square lattice is applicable to $H^{\text{even}}$ as well: the ground state of $H^{\text{even}}$ is in the flux sector where all the eigenvalues of $\phi^{\text{even}}_{p,\vec{r}} = \phi^{\text{even}}_{p',\vec{r}} = -1$ ($\pi$-flux sector). So that we can choose $D_{\vec{r}} = 1$ and $D'_{\vec{r}} = -1$ everywhere to obtain the energy dispersion of in the ground state sector,

$$\varepsilon^{\text{even}}_{a(b),\mu} = \pm 2 \left| J^{a(b)}_{1,\mu} + J^{a(b)}_{2,\mu} e^{-i\vec{k}\cdot\vec{n}_1} + J^{a(b)}_{3,\mu} e^{-i\vec{k}\cdot(\vec{n}_1+\vec{n}_2)} - J^{a(b)}_{4,\mu} e^{i\vec{k}\cdot\vec{n}_2} \right|. \tag{16}$$

The energy dispersion for each band in Eq. (16) will be gapped if one of the four $|J^{a(b)_{\lambda,\mu}}|$ is greater than the sum of the remaining three, while will be gapless with two Dirac cones otherwise. A model characterized by an even Chern number $\nu = 2q-2$ can be obtained from $H^{\text{even}}$ by adding a perturbation $H'^{\text{even}} = H'_a + H'_b$ to open gaps in the Dirac spectra, similar as what we did for $\nu = 2q-1$. Here $H'_a$ and $H'_b$ are defined in Eqs. (11). In terms of Majorana fermions, $H'^{\text{even}}$ reads

$$H'^{\text{even}} = i\kappa \sum_{\vec{r}} \Big( \sum_{\mu} \big( \gamma_{\vec{r},A,\mu}\gamma_{\vec{r}-\vec{n}_1,A,\mu} - \gamma_{\vec{r},B,\mu}\gamma_{\vec{r}-\vec{n}_1,B,\mu} \big) + \sum_{\mu}' \big( \eta_{\vec{r},A,\mu}\eta_{\vec{r}-\vec{n}_1,A,\mu} - \eta_{\vec{r},B,\mu}\eta_{\vec{r}-\vec{n}_1,B,\mu} \big) \Big).$$

The perturbed system $H^{\text{even}} + H'^{\text{even}}$ remains exactly solvable since $H'^{\text{even}}$ commutes with $\hat{D}_{\vec{r}}$ and $\hat{D}'_{\vec{r}}$. The energy dispersion of $H^{\text{even}} + H'^{\text{even}}$ takes a form of

$$\varepsilon^{\text{even}}_{a(b),\mu}(\vec{k}) = \pm 2 \sqrt{J^2 \left| 1 + e^{-i\vec{k}\cdot\vec{n}_1} + e^{-i\vec{k}\cdot(\vec{n}_1+\vec{n}_2)} - e^{-i\vec{k}\cdot\vec{n}_2} \right|^2 + \Delta^2(\vec{k})}, \tag{17}$$

where we have set all the coupling constants in the Hamiltonian $H^{\text{even}}$ to be a single value $J$ for simplicity.

With the help of parallel analyse to those for $\nu = 2q-1$, we are able to obtain the $\nu = 2q-2$ topologically ordered phase that respects the SO($2q-2$) symmetry, and other $\nu = -2q+2, \cdots, 2q-2$ topological phases that break the SO($2q-2$) symmetry.

## 3.3 Alternative construction for even Chern numbers

In this subsection, we would like to provide alternative construction for $\nu = 2q-2$ model, where each unit cell consists of $2q-1$ ($q \geq 2$) rather than $2q$ sites. In this approach, we define our models on a $L_1 \times L_2 \times (2q-1)$ 2D lattice as plotted in Fig. 7 (a), on which each site is labeled by $(\vec{r}, \mu)$, i.e., the unit cell vector $\vec{r}$ and a basis index $\mu$ ($\mu = 1, 2, \ldots, 2q-1$). The Hamiltonian $\tilde{H}^{\text{even}}$ takes a three-part form:

$$\tilde{H}^{\text{even}} = \tilde{H}^{\text{even}}_a + \tilde{H}^{\text{even}}_b + \tilde{H}^{\text{even}}_c, \tag{18a}$$

with

$$\tilde{H}^{\text{even}}_a = (-1)^q \sum_{\vec{r}} \widetilde{\sum_{\mu}} \Bigg( \tilde{J}^a_{1,\mu} \sigma^x_{\vec{r},\mu} \Bigg( \prod_{\mu+1 \leqslant \rho \leqslant q-2+\mu} \sigma^z_{\vec{r},\rho} \Bigg) \sigma^y_{\vec{r},q-1+\mu}$$

$$+ \tilde{J}^a_{2,\mu} \sigma^x_{\vec{r}-\vec{n}_1,q-1+\mu} \Bigg( \prod_{q+\mu \leqslant \rho \leqslant 2q-1} \sigma^z_{\vec{r}-\vec{n}_1,\rho} \Bigg) \Bigg( \prod_{1 \leqslant \rho \leqslant \mu-1} \sigma^z_{\vec{r},\rho} \Bigg) \sigma^y_{\vec{r},\mu} \tag{18b}$$

$$- \tilde{J}^a_{3,\mu} \sigma^x_{\vec{r}-\vec{n}_1-\vec{n}_2,q-1+\mu} \Bigg( \prod_{q+\mu \leqslant \rho \leqslant 2q-2} \sigma^z_{\vec{r}-\vec{n}_1-\vec{n}_2,\rho} \Bigg) \sigma^x_{\vec{r}-\vec{n}_1-\vec{n}_2,2q-1} \sigma^y_{\vec{r},1} \Bigg( \prod_{2 \leqslant \rho \leqslant \mu-1} \sigma^z_{\vec{r},\rho} \Bigg) \sigma^y_{\vec{r},\mu} \Bigg),$$

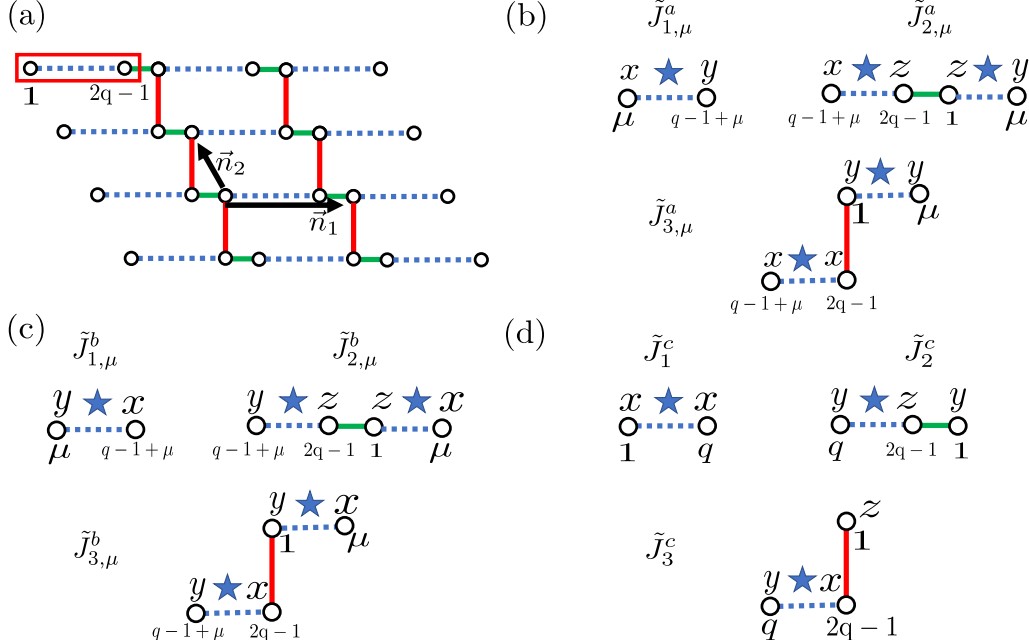

Figure 7: (a) A 2D lattice on which the Hamiltonian (18) is defined. Each unit cell consists of $2q-1$ sites and dashed lines represent the abbreviated $2q-3$ sites between the sites $\mu = 1$ and $\mu = 2q-1$. Each unit cell is labeled by $\vec{r} = l_1\vec{n}_1 + l_2\vec{n}_2$ ($l_1$, $l_2$ $\in \mathbb{N}_+$), where $\vec{n}_1$ and $\vec{n}_2$ are the primitive vectors for the lattice. (b), (c) and (d) denotes the terms in $\tilde{H}_a^{\text{even}}$, $\tilde{H}_b^{\text{even}}$ and $\tilde{H}_c^{\text{even}}$, respectively. Blue stars represent the product of $\sigma^z$'s on corresponding abbreviated sites.

$$
\begin{aligned}
\tilde{H}_b^{\text{even}} = (-1)^{q+1} \sum_{\vec{r}} \widetilde{\sum_{\mu}}' \Bigg( & \tilde{J}_{1,\mu}^b \sigma_{\vec{r},\mu}^y \left( \prod_{\mu+1\leqslant\rho\leqslant q-2+\mu} \sigma_{\vec{r},\rho}^z \right) \sigma_{\vec{r},q-1+\mu}^x \\
& + \tilde{J}_{2,\mu}^b \sigma_{\vec{r}-\vec{n}_1,\, q-1+\mu}^y \left( \prod_{q+\mu\leqslant\rho\leqslant 2q-1} \sigma_{\vec{r}-\vec{n}_1,\rho}^z \right) \left( \prod_{1\leqslant\rho\leqslant\mu-1} \sigma_{\vec{r},\rho}^z \right) \sigma_{\vec{r},\mu}^x \\
& - \tilde{J}_{3,\mu}^b \sigma_{\vec{r}-\vec{n}_1-\vec{n}_2,\, q-1+\mu}^y \left( \prod_{q+\mu\leqslant\rho\leqslant 2q-2} \sigma_{\vec{r}-\vec{n}_1-\vec{n}_2,\rho}^z \right) \sigma_{\vec{r}-\vec{n}_1-\vec{n}_2,\, 2q-1}^x \sigma_{\vec{r},1}^y \left( \prod_{2\leqslant\rho\leqslant\mu-1} \sigma_{\vec{r},\rho}^z \right) \sigma_{\vec{r},\mu}^x \Bigg),
\end{aligned}
\tag{18c}
$$

$$
\begin{aligned}
\tilde{H}_c^{\text{even}} = (-1)^q \sum_{\vec{r}} \Bigg( & -\tilde{J}_1^c \sigma_{\vec{r},1}^x \left( \prod_{2\leqslant\rho\leqslant q-1} \sigma_{\vec{r},\rho}^z \right) \sigma_{\vec{r},q}^x + \tilde{J}_2^c \sigma_{\vec{r}-\vec{n}_1,q}^y \left( \prod_{q+1\leqslant\rho\leqslant 2q-1} \sigma_{\vec{r}-\vec{n}_1,\rho}^z \right) \sigma_{\vec{r},1}^y \\
& + \tilde{J}_3^c \sigma_{\vec{r}-\vec{n}_1-\vec{n}_2,q}^y \left( \prod_{q+1\leqslant\rho\leqslant 2q-2} \sigma_{\vec{r}-\vec{n}_1-\vec{n}_2,\rho}^z \right) \sigma_{\vec{r}-\vec{n}_1-\vec{n}_2,\, 2q-1}^x \sigma_{\vec{r},1}^z \Bigg),
\end{aligned}
\tag{18d}
$$

where $\tilde{J}_{\lambda,\mu}^{a(b)}$ and $\tilde{J}_\lambda^c$ ($\lambda = 1, 2, 3$) are coupling constants, and the summations $\widetilde{\sum}_\mu = \sum_{\mu=2}^q$ and $\widetilde{\sum}'_\mu = \sum_{\mu=2}^{q-1}$, respectively.

As illustrated in Fig. 8, a flux operator $\hat{\phi}_{p,\vec{r}}^{\text{even}}$ can be defined as follows,

$$
\hat{\phi}_{p,\vec{r}}^{\text{even}} = -\sigma_{\vec{r},2q-1}^y \sigma_{\vec{r}+\vec{n}_1+\vec{n}_2,1}^y \left( \prod_{2\leqslant\rho\leqslant 2q-1} \sigma_{\vec{r}+\vec{n}_1+\vec{n}_2,\rho}^z \right) \sigma_{\vec{r}+2\vec{n}_1+\vec{n}_2,1}^x \sigma_{\vec{r}+\vec{n}_1,2q-1}^x \left( \prod_{1\leqslant\rho\leqslant 2q-2} \sigma_{\vec{r}+\vec{n}_1,\rho}^z \right). \tag{19}
$$

Similarly, all the flux operators $\hat{\phi}_{p,\vec{r}}^{\text{even}}$ commute with each other and with the Hamiltonian (18),

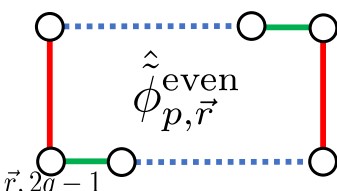

Figure 8: A plaquette where the flux operator $\hat{\tilde{\phi}}_{p,\vec{r}}^{\text{even}}$ in Eq. (19) is defined.

and $\left(\hat{\tilde{\phi}}_{p,\vec{r}}^{\text{even}}\right)^2 = 1$. The eigenvalues of $\hat{\tilde{\phi}}_{p,\vec{r}}^{\text{even}}$ compose a set of good quantum numbers $\left\{\tilde{\phi}_{p,\vec{r}}^{\text{even}}\right\}$, where $\tilde{\phi}_{p,\vec{r}}^{\text{even}} = \pm 1$.

Define the order of sites as follows: for two sites $l$ and $m$, (1) if $l_2 < m_2$, then $l < m$; (2) if $l_2 = m_2$ and $l_1 < m_1$, then $l < m$; (3) if $l_2 = m_2$, $l_1 = m_1$, and $\mu_l < \mu_m$, then $l < m$. We can perform the Jordan-Wigner transformation to fermionize the spin-1/2 model Hamiltonian $\tilde{H}^{\text{even}}$, resulting in,

$$\tilde{H}_a^{\text{even}} = i \sum_{\vec{r}} \widetilde{\sum_{\mu}} \left( \tilde{J}_{1,\mu}^a \gamma_{\vec{r},\mu} \gamma_{\vec{r},q-1+\mu} + \tilde{J}_{2,\mu}^a \gamma_{\vec{r},\mu} \gamma_{\vec{r}-\vec{n}_1,q-1+\mu} + \tilde{J}_{3,\mu}^a \hat{\tilde{D}}_{\vec{r}-\vec{n}_1-\vec{n}_2} \gamma_{\vec{r},\mu} \gamma_{\vec{r}-\vec{n}_1-\vec{n}_2,q-1+\mu} \right),$$

(20a)

$$\tilde{H}_b^{\text{even}} = i \sum_{\vec{r}} \widetilde{\sum_{\mu}}' \left( \tilde{J}_{1,\mu}^b \eta_{\vec{r},\mu} \eta_{\vec{r},q-1+\mu} + \tilde{J}_{2,\mu}^b \eta_{\vec{r},\mu} \eta_{\vec{r}-\vec{n}_1,q-1+\mu} + \tilde{J}_{3,\mu}^b \hat{\tilde{D}}_{\vec{r}-\vec{n}_1-\vec{n}_2} \eta_{\vec{r},\mu} \eta_{\vec{r}-\vec{n}_1-\vec{n}_2,q-1+\mu} \right), \quad \text{(20b)}$$

$$\tilde{H}_c^{\text{even}} = i \sum_{\vec{r}} \left( \tilde{J}_1^c \gamma_{\vec{r},1} \eta_{\vec{r},q} + \tilde{J}_2^c \gamma_{\vec{r},1} \eta_{\vec{r}-\vec{n}_1,q} + \tilde{J}_3^c \hat{\tilde{D}}_{\vec{r}-\vec{n}_1-\vec{n}_2} \gamma_{\vec{r},1} \eta_{\vec{r}-\vec{n}_1-\vec{n}_2,q} \right), \quad \text{(20c)}$$

where the complex fermions $f_m^\dagger$ have been decomposed into two Majorana fermions in the same way: $\eta_m = f_m^\dagger + f_m$ and $\gamma_m = i(f_m^\dagger - f_m)$. Here two Majorana fermions $\eta_{\vec{r},1}$ and $\eta_{\vec{r},q}$ constitute the gauge field $\hat{\tilde{D}}_{\vec{r}} = i\eta_{\vec{r},2q-1}\eta_{\vec{r}+\vec{n}_1+\vec{n}_2,1}$, which commutes with each other and with the Hamiltonian $\tilde{H}^{\text{even}}$, and $\hat{\tilde{D}}_{\vec{r}}^2 = 1$. The corresponding $\mathbb{Z}_2$ flux reads $\hat{\tilde{\phi}}_{p,\vec{r}}^{\text{even}} = \hat{\tilde{D}}_{\vec{r}} \hat{\tilde{D}}_{\vec{r}+\vec{n}_1}$.

Note that $\tilde{H}_a^{\text{even}}$, $\tilde{H}_b^{\text{even}}$ and $\tilde{H}_c^{\text{even}}$ are composed of $q-1$ [$\gamma_{\vec{r}\mu}$ and $\gamma_{\vec{r},q-1+\mu}$ ($\mu = 2, 3, \ldots, q$)], $q-2$ [$\eta_{\vec{r},\mu}$ and $\eta_{\vec{r},q-1+\mu}$ ($\mu = 2, 3, \ldots, q-1$)] and 1 [$\gamma_{\vec{r},1}$ and $\eta_{\vec{r},q}$] species of itinerant Majorana fermions, respectively. All the $2q-2$ species of itinerant Majorana fermions are decoupled to each other and coupled to the same $\mathbb{Z}_2$ gauge field. Since the pairing form of each species of itinerant Majorana fermions takes the same form as that in Eq. (4), the ground state of $\tilde{H}^{\text{even}}$ must be in the zero flux sector on which $\tilde{\phi}_{p,\vec{r}}^{\text{even}} = 1$ everywhere. The energy dispersion in the ground state sector reads

$$\tilde{\varepsilon}_{a(b),\mu}^{\text{even}}(\vec{k}) = \pm 2 \left| \tilde{J}_{1,\mu}^{a(b)} + \tilde{J}_{2,\mu} e^{-i\vec{k}\cdot\vec{n}_1} + \tilde{J}_{3,\mu}^{a(b)} e^{-i\vec{k}\cdot(\vec{n}_1+\vec{n}_2)} \right|, \quad \text{(21a)}$$

$$\tilde{\varepsilon}_c^{\text{even}}(\vec{k}) = \pm 2 \left| \tilde{J}_1^c + \tilde{J}_2^c e^{-i\vec{k}\cdot\vec{n}_1} + \tilde{J}_3^c e^{-i\vec{k}\cdot(\vec{n}_1+\vec{n}_2)} \right|. \quad \text{(21b)}$$

To obtain a model characterized by an even Chern number $\nu = 2q-2$, we tune the coupling constants in the Hamiltonian (18) to set all the $2q-2$ filled bands in Eqs. (21) gapless at first. Then we introduce the following perturbation Hamiltonian,

$$\tilde{H}'^{\text{even}} = \tilde{H}_a' + \tilde{H}_b' + \tilde{H}_c', \quad \text{(22a)}$$

which is made of three parts,

$$
\begin{aligned}
\tilde{H}'_a = \kappa \sum_{\vec{r}} \widetilde{\sum_{\mu}} \Bigg( &- \sigma^x_{\vec{r}-\vec{n}_1,\mu} \left( \prod_{\mu+1\leqslant\rho\leqslant 2q-1} \sigma^z_{\vec{r}-\vec{n}_1,\rho} \right) \left( \prod_{1\leqslant\rho\leqslant\mu-1} \sigma^z_{\vec{r},\rho} \right) \sigma^y_{\vec{r},\mu} \\
&+ \sigma^x_{\vec{r}-\vec{n}_1,q-1+\mu} \left( \prod_{q+\mu\leqslant\rho\leqslant 2q-1} \sigma^z_{\vec{r}-\vec{n}_1,\rho} \right) \left( \prod_{1\leqslant\rho\leqslant q-2+\mu} \sigma^z_{\vec{r},\rho} \right) \sigma^y_{\vec{r},q-1+\mu} \Bigg),
\end{aligned}
\tag{22b}
$$

$$
\begin{aligned}
\tilde{H}'_b = \kappa \sum_{\vec{r}} \widetilde{\sum_{\mu}}' \Bigg( &\sigma^y_{\vec{r}-\vec{n}_1\,\mu} \left( \prod_{\mu+1\leqslant\rho\leqslant 2q-1} \sigma^z_{\vec{r}-\vec{n}_1,\rho} \right) \left( \prod_{1\leqslant\rho\leqslant\mu-1} \sigma^z_{\vec{r},\rho} \right) \sigma^x_{\vec{r}\,\mu} \\
&- \sigma^y_{\vec{r}-\vec{n}_1\,q-1+\mu} \left( \prod_{q+\mu\leqslant\rho\leqslant 2q-1} \sigma^z_{\vec{r}-\vec{n}_1,\rho} \right) \left( \prod_{1\leqslant\rho\leqslant q-2+\mu} \sigma^z_{\vec{r},\rho} \right) \sigma^x_{\vec{r}\,q-1+\mu} \Bigg),
\end{aligned}
\tag{22c}
$$

$$
\begin{aligned}
\tilde{H}'_c = -\kappa \sum_{\vec{r}} \Bigg( &\sigma^x_{\vec{r}-\vec{n}_1,1} \left( \prod_{2\leqslant\rho\leqslant 2q-1} \sigma^z_{\vec{r}-\vec{n}_1,\rho} \right) \sigma^y_{\vec{r},1} \\
&+ \sigma^y_{\vec{r}-\vec{n}_1,q} \left( \prod_{q+1+\mu\leqslant\rho\leqslant 2q-1} \sigma^z_{\vec{r}-\vec{n}_1,\rho} \right) \left( \prod_{1\leqslant\rho\leqslant q-1} \sigma^z_{\vec{r},\rho} \right) \sigma^x_{\vec{r},q} \Bigg).
\end{aligned}
\tag{22d}
$$

The Jordan-Wigner transformation will fermionize the Hamiltonian $\tilde{H}'^{\text{even}}$ to be

$$
\begin{aligned}
\tilde{H}'^{\text{even}} = i\kappa \sum_{\vec{r}} \widetilde{\sum} \Bigg( &\sum_{\mu} \left( \gamma_{\vec{r},\mu}\gamma_{\vec{r}-\vec{n}_1,\mu} - \gamma_{\vec{r},q-1+\mu}\gamma_{\vec{r}-\vec{n}_1,q-1+\mu} \right) \\
&+ \widetilde{\sum_{\mu}}' \left( \eta_{\vec{r},\mu}\eta_{\vec{r}-\vec{n}_1,\mu} - \eta_{\vec{r},q-1+\mu}\eta_{\vec{r}-\vec{n}_1,q-1+\mu} \right) + \left( \gamma_{\vec{r},1}\gamma_{\vec{r}-\vec{n}_1,1} - \eta_{\vec{r},q}\eta_{\vec{r}-\vec{n}_1,q} \right) \Bigg).
\end{aligned}
$$

The perturbed system $\tilde{H}^{\text{even}} + \tilde{H}'^{\text{even}}$ remains exactly solvable, since $\tilde{H}'^{\text{even}}$ commutes with $\hat{\tilde{D}}_{\vec{r}}$. The perturbation $\tilde{H}'^{\text{even}}$ opens energy gaps in the spectra, yielding

$$
\tilde{\varepsilon}^{\text{odd}}_{a(b),\mu}(\vec{k}) = \tilde{\varepsilon}^{\text{odd}}_c(\vec{k}) = \pm 2\sqrt{J^2 \left| 1 + e^{-i\vec{k}\cdot\vec{n}_1} + e^{-i\vec{k}\cdot(\vec{n}_1+\vec{n}_2)} \right|^2 + \Delta^2(\vec{k})},
\tag{23}
$$

where $\Delta(\vec{k}) = 2\kappa \sin(\vec{k}\cdot\vec{n}_1)$ and we have set all the coupling constants equal $J$. Each filled band in Eq. (23) gives rise to a Chern number $\nu = \text{sgn}(\kappa)$, and we obtain the $\nu = 2q-2$ topologically ordered phase that respects the SO($2q-2$) symmetry, and other $\nu = -2q+2, \cdots, 2q-2$ topological phases that break the SO($2q-2$) symmetry, following the parallel analyse to those for $\nu = 2q-1$.

## 4 Topologically degenerate ground states

The topological order can manifest itself via the ground state degeneracy on a manifold with nonzero genus [2]. In this section, we study exactly solvable spin-1/2 models under PBC, which allows global $\mathbb{Z}_2$ fluxes along two directions. To do this, we shall treat the physical boundary condition for the Jordan-Wigner transformation properly [18, 40].

## 4.1 Periodic boundary condition, boundary terms, and local and global $\mathbb{Z}_2$ fluxes

Without loss of generality, we consider a $L_1 \times L_2 \times 2q$ lattice with (PBC) along both $\vec{n}_1$ and $\vec{n}_2$ directions. To be simple, $L_1$ and $L_2$ will be chosen to be even numbers hereafter. Under PBC, there will appear additional boundary terms in Hamiltonian (7) and (13) [18]. In terms of Majorana fermions, these additional boundary terms read

$$(-1)^q J^a_{2,\mu} \sigma^y_{(L_1,l_2),B,\mu} \left( \prod_{\mu+1\leqslant\rho\leqslant q} \sigma^z_{(L_1,l_2),B,\rho} \right) \left( \prod_{1\leqslant\rho\leqslant\mu-1} \sigma^z_{(1,l_2),A,\rho} \right) \sigma^y_{(1,l_2),A,\mu}$$
$$= i J^a_{2,\mu} \gamma_{(1,l_2),A,\mu} \gamma_{(L_1,l_2),B,\mu} \hat{F}_{l_2} ,$$

$$(-1)^q J^a_{3,\mu} \sigma^y_{(L_1,l_2),B,\mu} \left( \prod_{\mu+1\leqslant\rho\leqslant q-1} \sigma^z_{(L_1,l_2),B,\rho} \right) \sigma^y_{(L_1,l_2),B,q} \sigma^y_{(1,l_2+1),A,1} \left( \prod_{2\leqslant\rho\leqslant\mu-1} \sigma^z_{(1,l_2+1),A,\rho} \right) \sigma^y_{(1,l_2+1),A,\mu}$$
$$= i J^a_{3,\mu} \hat{D}_{(L_1,l_2)} \gamma_{(1,l_2+1),A,\mu} \gamma_{(L_1,l_2),B,\mu} \hat{F}_{l_2} ,$$

$$(-1)^{q+1} J^b_{2,\mu} \sigma^x_{(L_1,l_2),B,\mu} \left( \prod_{\mu+1\leqslant\rho\leqslant q} \sigma^z_{(L_1,l_2),B,\rho} \right) \left( \prod_{1\leqslant\rho\leqslant\mu-1} \sigma^z_{(1,l_2),A,\rho} \right) \sigma^x_{(1,l_2),A,\mu}$$
$$= i J^b_{2,\mu} \eta_{(1,l_2),A,\mu} \eta_{(L_1,l_2),B,\mu} \hat{F}_{l_2} ,$$

$$(-1)^{q+1} J^b_{3,\mu} \sigma^x_{(L_1,l_2),B,\mu} \left( \prod_{\mu+1\leqslant\rho\leqslant q-1} \sigma^z_{(L_1,l_2),B,\rho} \right) \sigma^y_{(L_1,l_2),B,q} \sigma^y_{(1,l_2+1),A,1} \left( \prod_{2\leqslant\rho\leqslant\mu-1} \sigma^z_{(1,l_2+1),A,\rho} \right) \sigma^x_{(1,l_2+1),A,\mu}$$
$$= i J^b_{3,\mu} \hat{D}_{(L_1,l_2)} \eta_{(1,l_2+1),A,\mu} \eta_{(L_1,l_2),B,\mu} \hat{F}_{l_2} ,$$

$$J^c_2 \sigma^x_{(L_1,l_2),B,1} \left( \prod_{2\leqslant\rho\leqslant q} \sigma^z_{(L_1,l_2),B,\rho} \right) \left( \prod_{1\leqslant\rho\leqslant q-1} \sigma^z_{(1,l_2),A,\rho} \right) \sigma^x_{(1,l_2),A,q}$$
$$= i J^c_2 \eta_{(1,l_2),A,q} \eta_{(L_1,l_2),B,1} \hat{F}_{l_2} ,$$

$$J^c_3 \sigma^x_{(L_1,l_2),B,1} \left( \prod_{2\leqslant\rho\leqslant q-1} \sigma^z_{(L_1,l_2),B,\rho} \right) \sigma^y_{(L_1,l_2),B,q} \sigma^y_{(1,l_2+1),A,1} \left( \prod_{2\leqslant\rho\leqslant q-1} \sigma^z_{(1,l_2+1),A,\rho} \right) \sigma^x_{(1,l_2+1),A,q}$$
$$= i J^c_3 \hat{D}_{(L_1,l_2)} \eta_{(1,l_2+1),A,q} \eta_{(L_1,l_2),B,1} \hat{F}_{l_2} ,$$

where $(l_1, l_2)$ denotes the unit cell $\vec{r} = l_1\vec{n}_1 + l_2\vec{n}_2$, $\hat{F}_{l_2} = e^{i\pi\hat{N}_{l_2}}$ and $\hat{N}_{l_2} = \sum_{l_1,\beta,\mu} \hat{n}_{(l_1,l_2),\beta,\mu}$ are fermion parity and occupation number in the $l_2$-th row respectively. Note the coupling constant $J^{a(b)}_{4,\mu}$ is absent in these additional boundary terms. Thus every boundary term in $H^{\text{even}}$ is a boundary term in $H^{\text{odd}}$ too. The fermion parity $\hat{F}_{l_2}$ will also appear in the flux operators on the edge,

$$\hat{\phi}^{\text{odd}}_{p,(L_1,l_2)} = \hat{D}_{(L_1,l_2)} \hat{D}_{(1,l_2)} \hat{F}_{l_2} ,$$
$$\hat{\phi}^{\text{odd}}_{p,(L_1-1,l_2)} = \hat{D}_{(L_1-1,l_2)} \hat{D}_{(1,l_2)} \hat{F}_{l_2+1} ,$$
$$\hat{\phi}^{\text{even}}_{p,(L_1,l_2)} = \hat{D}_{(L_1,l_2)} \hat{D}'_{(1,l_2)} \hat{F}_{l_2} ,$$
$$\hat{\phi}^{\text{even}}_{p',(L_1,l_2)} = \hat{D}_{(L_1,l_2)} \hat{D}_{(L_1,l_2)} \hat{F}_{l_2+1} .$$

In addition to the local flux operators, we can define two extra $\mathbb{Z}_2$ global flux operators:

$$\hat{\Phi}_1 = \hat{F}_{l_2=1},$$
$$\hat{\Phi}_2 = \prod_{l_2} \hat{D}_{(1,l_2)}.$$

These two global fluxes commute with each other and with $\hat{\phi}_{p,\vec{r}}^{\text{odd}}$, $\hat{\phi}_{p,\vec{r}}^{\text{even}}$ and $\hat{\phi}_{p',\vec{r}}^{\text{even}}$ as well as $H^{\text{odd}}$ and $H^{\text{even}}$ under PBC. It is easy to see that $\hat{\Phi}_{1(2)}^2 = 1$, and the corresponding eigenvalue is $\Phi_{1(2)} = \pm 1$. Thus we can divide the total Hilbert space of $H^{\text{odd}}$ ($H^{\text{even}}$) into subspaces according to the sets of eigenvalues $\left\{\phi_{p,\vec{r}}^{\text{odd}}, \Phi_1, \Phi_2\right\}$ or $\left\{\phi_{p,\vec{r}}^{\text{even}}, \phi_{p',\vec{r}}^{\text{even}}, \Phi_1, \Phi_2\right\}$.

## 4.2 Topologically degenerate ground states on a torus

To study the topological degeneracy of spin-1/2 models, let us count the degrees of freedom at first. For a spin-1/2 model defined on an $L_1 \times L_2 \times 2q$ lattice, there are $2^{2qL_1L_2}$ physical spin states. On the other hand, we consider the degrees of freedom arising from fermions that are subject to a constraint $\Pi_{\vec{r}}\phi_{p,\vec{r}}^{\text{odd}} = 1$ or $\Pi_{\vec{r}}\phi_{p,\vec{r}}^{\text{even}}\phi_{p',\vec{r}}^{\text{even}} = 1$ under the PBC on a torus: (1) for a $\nu = 2q - 1$ model, there are $(L_1L_2 + 2)$ $\mathbb{Z}_2$ fluxes and $(4q-2)L_1L_2$ itinerant Majorana fermions, giving rise to $2^{L_1L_2+2} \times \frac{1}{2} \times 2^{(2q-1)L_1L_2} = 2^{2qL_1L_2+1}$ states; (2) for a $\nu = 2q - 2$ model, there are $(2L_1L_2 + 2)$ $\mathbb{Z}_2$ fluxes and $(4q-4)L_1L_2$ itinerant Majorana fermions, resulting in $2^{2L_1L_2+2} \times \frac{1}{2} \times 2^{(2q-2)L_1L_2} = 2^{2qL_1L_2+1}$ states as well. Therefore the degrees of freedom in the Fock space composed of Majorana fermions have been enlarged by a factor of two. This unphysical redundancy can be removed by a projection $\hat{P} = (1+F\hat{F})/2$ [40], where $\hat{F} = \prod_{l_2} \hat{F}_{l_2}$ is the total fermion number parity and $F$ is its eigenvalue for a given set of $\left\{\phi_{p,\vec{r}}^{\text{odd}}, \Phi_1, \Phi_2\right\}$ $\left(\left\{\phi_{p,\vec{r}}^{\text{even}}, \phi_{p',\vec{r}}^{\text{even}}, \Phi_1, \Phi_2\right\}\right)$ for $H^{\text{odd}}$ ($H^{\text{even}}$).

In the Majorana fermion representation, the ground states of $H^{\text{odd}}$ and $H^{\text{even}}$ lie in the subspace where the eigenvalues of local flux operators have been determined. The undetermined eigenvalues of global flux operators $\hat{\Phi}_1$ and $\hat{\Phi}_2$ leads to four-fold topologically degenerate ground states $|\Psi_G(\Phi_1, \Phi_2)\rangle$ on the torus, characterized by $\Phi_1 = \pm 1$ and $\Phi_2 = \pm 1$. Indeed, $\Phi_{1(2)} = 1$ and $\Phi_{1(2)} = -1$ give rise to periodic and anti-periodic boundary conditions for itinerant Majorana fermions along the $\vec{n}_{1(2)}$ direction, respectively. For $\Phi_{1(2)} = 1$, the $\vec{n}_{1(2)}$-component of the wave vector $\vec{k}$ is determined by $k_{1(2)} \equiv \vec{k} \cdot \vec{n}_{1(2)} = 0, \pm 2\pi/L_{1(2)}, \pm 4\pi/L_{1(2)}, \cdots, \pm(L_{1(2)}-2)\pi/L_{1(2)}, \pi$; while for $\Phi_{1(2)} = -1$, the corresponding value is given by $k_{1(2)} = \pm \pi/L_{1(2)}, \pm 3\pi/L_{1(2)}, \cdots, \pm(L_{1(2)}-1)\pi/L_{1(2)}$. On the other hand, in the gapless phase of the original Kitaev model, the quadratic form in Eq. (4) leads to a $p$-wave pairing term around each Dirac cone. Moreover, a TRS breaking perturbation will result in a $p \pm ip$ pairing term and open an energy gap at the Dirac cone. Thus, when $\Phi_1 = \Phi_2 = 1$, the $p \pm ip$ pairing term vanishes at $k_1 = k_2 = 0$, which gives rise to unpaired fermions at the Fermi level and a sign change of the presumed value $F$ in each filled band, resulting in $(1 + F\hat{F})|\Psi_G(\Phi_1 = 1, \Phi_2 = 1)\rangle = 0$ [40]. By contrast, the fermions are fully paired in other three topological sectors, on which either $\Phi_1 = -1$ or $\Phi_2 = -1$. Note that the fermions are fully paired in the gapped $\nu = 0$ phase too, whatever $\Phi_1$ and $\Phi_2$ are.

As mentioned before, the models in Eqs. (9) and (15) can be viewed as $\nu$ copies of itinerant Majorana fermions in Eq. (4) that are coupled to a single static $\mathbb{Z}_2$ gauge field. When there are $\nu$ topologically non-trivial (i.e., $p \pm ip$ pairing) bands filled, an extra sign $(-1)^\nu$ will appear in the projector, yielding $(1 + F\hat{F})|\Psi_G(\Phi_1 = 1, \Phi_2 = 1)\rangle = [1 + (-1)^\nu]|\Psi_G(\Phi_1 = 1, \Phi_2 = 1)\rangle$. Consequently, the fermionic ground state with $\Phi_1 = \Phi_2 = 1$ will be eliminated by the projection $\hat{P}$, when $\nu$ is an odd number. Then we draw the conclusion that the physical ground states of $H^{\text{odd}}$ and $H^{\text{even}}$ are three- and four-fold topologically degenerate, respectively.

Table 1: xactly solvable models toward Kitaev's sixteen-fold way: comparison between our models ($H^{\text{odd}}$, $H^{\text{even}}$, and $\tilde{H}^{\text{even}}$) and those presented in previous works [36–39].

| | lattice | Chern number $\mathcal{C} = \nu$ | vortex sector | local degrees of freedom |
|---|---|---|---|---|
| Ref. [36] | square-octagon lattice | $\nu = 0, \pm 1, \pm 2, \pm 3, \pm 4$ | zero-flux | a spin-$\frac{1}{2}$ per site |
| Ref. [37] | honeycomb lattice | $\nu = 0, \pm 1, \pm 2, \pm 3, \pm 4, \pm 8$ | ground state: fractional-flux (or 1-flux) | a spin-$\frac{1}{2}$ per site |
| Ref. [38] | honeycomb lattice | $\nu = 0, \pm 1, \pm 2, \pm 3,$ $\pm 4, \pm 5, \pm 6, \pm 8$ | triangular vortex configurations and their dual | a spin-$\frac{1}{2}$ per site |
| Ref. [39] | honeycomb (square) lattice for odd (even) $\nu$ | $\nu = 2q - 1$ or $\nu = 2q - 2$ | ground state: zero-flux ($\pi$-flux) for odd (even) $\nu$ | $(2q + 1)$ $2^q$-dimensional $\Gamma$ matrices per site |
| $H^{\text{odd}}$ | brick-wall lattice: $2q$ sites per unit cell | $\nu = 2q - 1$ | ground state: zero-flux | a spin-$\frac{1}{2}$ per site |
| $H^{\text{even}}$ | brick-wall lattice: $2q$ sites per unit cell | $\nu = 2q - 2$ | ground state: $\pi$-flux | a spin-$\frac{1}{2}$ per site |
| $\tilde{H}^{\text{even}}$ | brick-wall lattice: $2q - 1$ sites per unit cell | $\nu = 2q - 2$ | ground state: zero-flux | a spin-$\frac{1}{2}$ per site |

# 5 Conclusions and discussions

In summary, we proposed a family of 2D quantum $S = 1/2$ spin models to realize Kitaev's sixteen-fold way of anyon theories. With the help of Jordan-Wigner transformation, all these spin models can be fermionized and mapped to quadratic form of Majorana fermions, thereby are exactly solvable. These exact solutions allow us to study the ground state degeneracy of these models on a torus. It turns out that the ground states are three- (four-) fold topologically degenerate, when the total Chern number $\nu$ of Majorana fermion bands is an odd (even) number.

For better understanding of our exactly solvable models on 2D brick-wall lattices, it is worthwhile to compare them with existing models hosting the sixteen Kitaev topological orders. In Table 1, we list some exactly solvable models in literature [36–39], and compare them with ours.

(i) We would like to point out the close relation between our $\nu = 2$ model and the spin-3/2 Yao-Zhang-Kivelson (YZK) model [32] that host algebraic spin liquid states. The spin-1/2 Hamiltonian $H^{\text{even}}$ in Eq. (13) can be fermionized via the Jordan-Wigner transformation, resulting in Eq. (15). When $\nu = 2$ (or $q = 2$), the latter Majorana fermion form can be explicitly written as follows,

$$H^{\nu=2} = i \sum_{\vec{r}} \sum_{\mu=1}^{2} \left( J^a_{1,\mu} \gamma_{\vec{r},A,\mu} \gamma_{\vec{r},B,\mu} + J^a_{2,\mu} \gamma_{\vec{r},A,\mu} \gamma_{\vec{r}-\vec{n}_1,B,\mu} \right.$$
$$\left. + J^a_{3,\mu} \hat{D}_{\vec{r}-\vec{n}_1-\vec{n}_2} \gamma_{\vec{r},A,\mu} \gamma_{\vec{r}-\vec{n}_1-\vec{n}_2,B,\mu} + J^a_{4,\mu} \hat{D}'_{\vec{r}} \gamma_{\vec{r},A,\mu} \gamma_{\vec{r}+\vec{n}_2,B,\mu} \right),$$

where two species of itinerant Majorana fermions $\gamma_{\vec{r},A,1}$ ($\gamma_{\vec{r},B,1}$) and $\gamma_{\vec{r},A,2}$ ($\gamma_{\vec{r},B,2}$) appear in the sublattice A(B). In order to compare with the YZK model, we add a new term $H^{\text{couple}}$ to

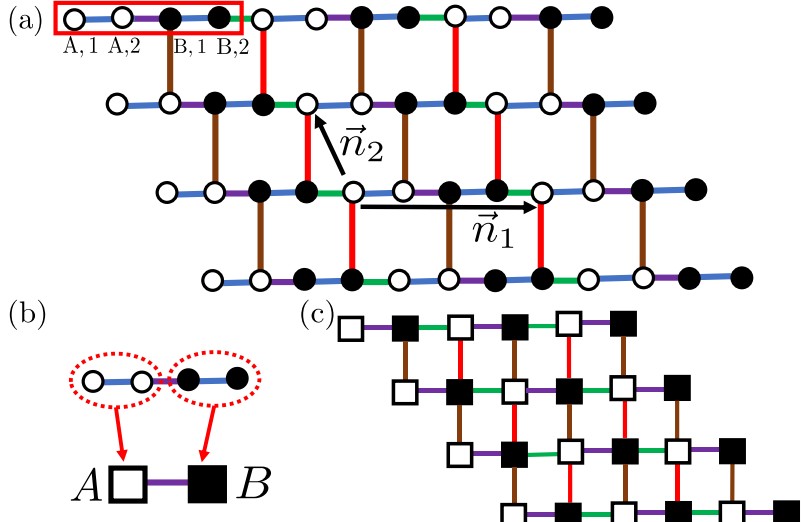

Figure 9: (a) The $q = 2$ version of the brick-wall lattice in Fig. 5 (a). (b) Combine two sites $A$, 1 and $A$, 2 ($B$, 1 and $B$, 2) into a single site $A$ ($B$). (c) The lattice in (a) is transformed into a square lattice via the site combination shown in (b). Here open and solid squares form two sublattices $A$ and $B$ respectively.

couple these two species of Majorana fermions with each other,

$$
\begin{aligned}
H^{\text{couple}} &= -\sum_{\vec{r}} J_5 \left( \sigma^x_{\vec{r},A,1} \sigma^y_{\vec{r},A,2} - \sigma^y_{\vec{r},B,1} \sigma^x_{\vec{r},B,2} \right) \\
&= -i \sum_{\vec{r}} J_5 \left( \gamma_{\vec{r},A,1} \gamma_{\vec{r},A,2} + \gamma_{\vec{r},B,1} \gamma_{\vec{r},B,2} \right).
\end{aligned}
\tag{24}
$$

Then we choose $J^a_{1,1} = J^a_{2,1} = J_x$, $J^a_{3,1} = J^a_{4,1} = J_y$, $J^b_{1,2} = J^b_{2,2} = J'_x$ and $J^b_{3,2} = J^b_{4,2} = J'_y$, such that the Hamiltonian $H^{\nu=2}_{\text{total}} \equiv H^{\nu=2} + H^{\text{couple}}$ becomes

$$
\begin{aligned}
H^{\nu=2}_{\text{total}} = i \sum_{\vec{r}} \Big( &J_x \gamma_{\vec{r},A,1} (\gamma_{\vec{r},B,1} + \gamma_{\vec{r}-\vec{n}_1,B,1}) + J_y \gamma_{\vec{r},A,1} (\hat{D}_{\vec{r}-\vec{n}_1-\vec{n}_2} \gamma_{\vec{r}-\vec{n}_1-\vec{n}_2,B,1} + \hat{D}'_{\vec{r}} \gamma_{\vec{r}+\vec{n}_2,B,1}) \\
&+ J'_x \gamma_{\vec{r},A,2} (\gamma_{\vec{r},B,2} + \gamma_{\vec{r}-\vec{n}_1,B,2}) + J'_y \gamma_{\vec{r},A,2} (\hat{D}_{\vec{r}-\vec{n}_1-\vec{n}_2} \gamma_{\vec{r}-\vec{n}_1-\vec{n}_2,B,2} + \hat{D}'_{\vec{r}} \gamma_{\vec{r}+\vec{n}_2,B,2}) \\
&- J_5 (\gamma_{\vec{r},A,1} \gamma_{\vec{r},A,2} + \gamma_{\vec{r},B,1} \gamma_{\vec{r},B,2}) \Big).
\end{aligned}
\tag{25}
$$

Note that $\hat{\phi}^{\text{even}}_{p,\vec{r}}$ and $\hat{\phi}^{\text{even}}_{p',\vec{r}}$ remain integrals of motion of $H^{\nu=2}_{\text{total}}$, and serve as static $\mathbb{Z}_2$ fluxes as well. It is straightforward to examine that the energy dispersion of Eq. (25) takes the same form as that in the YZK model [32]. Indeed, this equivalence can be understood as follows: As shown in Fig. 9, the $q = 2$ version of brick-wall lattice in Fig. 5 (a) can be transformed into a square lattice via coarse-graining [see Fig. 9 (b)]. The direct product of the Hilbert spaces of two $S = 1/2$ spins is associated with the four-dimensional representation of the Clifford algebra that composed of five $4 \times 4$ $\Gamma$ matrices and their commutators. On the other hand, the spin-3/2 representation of the SU(2) algebra are formulated in a four-dimensional Hilbert space too. Furthermore, all the five $4 \times 4$ $\Gamma$ matrices can be represented by symmetric bilinear combinations of the three SU(2) generators in the spin-3/2 representation. These allow us to establish the equivalence between these two models.

(ii) It is worth mentioning that Eqs. (9) is not the unique choice to divide the $4q - 2$ Majorana fermions into $2q - 1$ pairs. The way that we choose in Eqs. (9) keeps the index $\mu$ for each pair the same except for the pair, $\eta_{A,q}$ and $\eta_{B,1}$. This choice is convenient for our discussion, while leaves a long spin string operator of length $2q$ in $H^{\text{odd}}_c$. To reduce the length

of the longest spin string operator in $H^{\text{odd}}$, alternative pairing scheme is applicable: changing the pairing of $\eta$ Majorana fermions to $\eta_{A,\mu+1}\eta_{B,\mu}$ ($\mu = 2, 3, \ldots, q-1$). The corresponding spin-1/2 Hamiltonian contains up to $(q+2)$-spin interactions.

(iii) As mentioned in Section 3, some particular choice of the coupling constants will lead to an SO($|\nu|$) internal symmetry, which is absent in generic models. This enlarged symmetry will give rise to the same velocity of chiral Majorana fermions on the boundary, and featured entanglement spectra that are depicted by corresponding conformal field theories.

(iv) Finally, we would like to emphasize that our spin-1/2 models is easier to realize than those of higher spins via quantum simulation by various types of qubits. In particular, the long-range interacting terms in our models can be simulated, since the all-to-all interactions are feasible in cutting-edge quantum device techniques [43, 44].

# Acknowledgement

We would like to thank Hong-Hao Tu, Hui-Ke Jin, and Hong Yao for helpful discussions, and thank Yuan Wan for suggesting the word "lacing". This work is partially supported by National Key Research and Development Program of China (No. 2022YFA1403403), National Natural Science Foundation of China (No. 12274441, 12034004), the K. C. Wong Education Foundation (Grant No. GJTD-2020-01), and the Strategic Priority Research Program of Chinese Academy of Sciences (No. XDB28000000). J.-J. M. is supported by General Research Fund Grant No. 14302021 from Research Grants Council and Direct Grant No. 4053416 from The Chinese University of Hong Kong. We also thank the financial support from Innovation program for Quantum Science and Technology (Grant No. 2021ZD0302500)

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
