# Peer review of "Lacing topological orders in two dimensions: exactly solvable models for Kitaev's sixteen-fold way"

_SciPost Physics, doi:SciPost Phys. 14, 087 (2023)_

## Round 2 · Referee Report · Anonymous (Referee 1) · 2022-11-11

Strengths

1- A new construction of Kitaev-inspired lattice models which realize the 16-fold way of topological orders 2- The models constructed have $S=1/2$ elementary degrees of freedom and might thus be implemented in quantum simulators (cold-atom experiments) more easily 3- The authors provide a great amount of technical detail, making calculations reproducable

Weaknesses

1- The paper will be challenging to read for non-experts (as it is heavy on technical details) and lacks illustrations of some key concepts 2-Lacking details on realization: It is not clear to what extent the models are actually more "realistic" than previous constructions and how they can be realized in quantum simulators/quantum computers

Report

The paper is concerned with constructing bosonic microscopic lattice models which realize the 16-fold way of topological orders classified by Kitaev, which are obtained by coupling (gapped) Majorana fermions with Chern number $\nu$ to a Z2 gauge theory. Previous attempts have focussed on generating the required Majorana bands by flux engineering, or have started with enlarged local Hilbert spaces. In contrast, the authors of the work at hand are motivated by the Jordan-Wigner solution of Kitaev's honeycomb model and construct generalized brick-wall lattices of $S=1/2$ degrees of freedom. They appropriately gap out the itinerant Majorana fermions and demonstrate the topological degeneracies of the even/odd $\nu$ phases on the torus.

The work seems valid and is of value as it explores a new way of constructing microscopic models for the 16fold way. The paper is mostly well written and provides a lot technical details. However, I am not (yet) convinced the paper deserves publication in SciPost Physics given the acceptance criteria. Microscopic models for the 16fold way have been presented before so the work does not qualify as a breakthrough, and despite the author's comments, it appears that the model is not straightforwardly studied in quantum simulators (I wonder if the authors had any specific platforms/details concerning an AMO-implementation in mind?)

I also have two more detailed conceptual questions that the authors could address in their work:

1-The authors present a neat mapping which shows that the Yao-Zhang-Kivelson can be recovered within their construction. Indeed, I wonder if this can be generalized to all $\Gamma$-matrix models. It seems to me that grouping a sufficient number of $S=1/2$ will effectively give rise to one site with a larger Hilbert space on which the $\Gamma$-matrices act?

2-In the construction at hand, we have multiple Majorana fermion bands coupled to one Z2 gauge field. Kitaev's proposal for constructing higher-$\nu$ orders starts with multiple layers and then condenses interlayer anyons (see Sec. 10.9 in https://arxiv.org/abs/cond-mat/0506438v3) which binds the gauge fields together. It seems that the lacing construction presented by the authors may also be understood as the result of some layered construction. Could the authors comment on possible relations?

3-Can the authors comment if the models constructed are also solvable using Kitaev's fermionization procedure, or is the Jordan-Wigner construction required?

Requested changes

1-I highly recommend the authors add a few more illustrative figures concerning their construction. In particular to visualize the definition of the Jordan-Wigner path that is defined below page 2.

2-The comment on page 6 "it is natural to make $\nu$ copies of the $C=1$ lattice gauge theory" is confusing and does not actually match the authors construction, as it suggests that there are multiple copies of the Z2 gauge field. Instead, the construction presented only has one Z2 gauge theory (but it is interesting to ask if the model at hand can be recovered from literally taking $\nu$ copies of the $C=1$ model, see my question 2 above).

3-There are a few typos throughout the text, and undefined references on page 6 ("citeTu") and page 7 ("citeMiao1").

  • validity: high
  • significance: ok
  • originality: good
  • clarity: high
  • formatting: reasonable
  • grammar: reasonable

Author:  Yi Zhou  on 2022-12-15  [id 3137]

(in reply to Report 1 on 2022-11-11)

Weaknesses 1- The paper will be challenging to read for non-experts (as it is heavy on technical details) and lacks illustrations of some key concepts

Re: We have revised the paper in accordance with the reviewer’s suggestions. Hope this revised version will be more reader friendly.

2-Lacking details on realization: It is not clear to what extent the models are actually more "realistic" than previous constructions and how they can be realized in quantum simulators/quantum computers

Re: To be honest, we are not experts on quantum simulators/quantum computers. However, as a common sense, we believe that local two-state devices, i.e., qubits, are much easier to realize, in comparison with four (or more)-state devices in Josephson junctions or other quantum computation platforms.

Report The paper is concerned with constructing bosonic microscopic lattice models which realize the 16-fold way of topological orders classified by Kitaev, which are obtained by coupling (gapped) Majorana fermions with Chern number to a Z2 gauge theory. Previous attempts have focused on generating the required Majorana bands by flux engineering, or have started with enlarged local Hilbert spaces. In contrast, the authors of the work at hand are motivated by the Jordan-Wigner solution of Kitaev's honeycomb model and construct generalized brick-wall lattices of S=1/2 degrees of freedom. They appropriately gap out the itinerant Majorana fermions and demonstrate the topological degeneracies of the even/odd phases on the torus.

The work seems valid and is of value as it explores a new way of constructing microscopic models for the 16fold way. The paper is mostly well written and provides a lot technical details. However, I am not (yet) convinced the paper deserves publication in SciPost Physics given the acceptance criteria. Microscopic models for the 16fold way have been presented before so the work does not qualify as a breakthrough, and despite the author's comments, it appears that the model is not straightforwardly studied in quantum simulators (I wonder if the authors had any specific platforms/details concerning an AMO-implementation in mind?)

Re: We are grateful for the referee’s questions. We have considered all the constructive comments made by the referee, enclosed please find our point-to-point response.

I also have two more detailed conceptual questions that the authors could address in their work:

1-The authors present a neat mapping which shows that the Yao-Zhang-Kivelson can be recovered within their construction. Indeed, I wonder if this can be generalized to all \Gamma-matrix models. It seems to me that grouping a sufficient number of S=1/2 will effectively give rise to one site with a larger Hilbert space on which the \Gamma-matrices act?

Re: We thank the Referee for this question. Certainly, one can grouping S=1/2 spins to construct \Gamma-matrix models, in analogy with the coarse-graining scheme illustrated in Fig. 9 in this revised version. However, such a coarse-graining procedure is not necessary to be invertible. Therefore, the question whether an exactly solvable \Gamma-matrix model can be mapped to a spin-1/2 model is still open, unless an explicit invert-construction has been done as we did for the Yao-Zhang-Kivelson model.

\times\Gamma\Gamma\times\Gamma\Gamma\Gamma\Gamma\ \Gamma\ \Gamma 2-In the construction at hand, we have multiple Majorana fermion bands coupled to one Z2 gauge field. Kitaev's proposal for constructing higher-orders starts with multiple layers and then condenses interlayer anyons (see Sec. 10.9 in https://arxiv.org/abs/cond-mat/0506438v3) which binds the gauge fields together. It seems that the lacing construction presented by the authors may also be understood as the result of some layered construction. Could the authors comment on possible relations?

Re: Indeed, it has been shown in Ref. 31 that exactly solvable spin-1/2 models can be constructed in arbitrary dimensions by using the lacing method systematically. The basic idea is that a brick-wall lattice model can be made by coupling exactly solvable spin-1/2 chains via inter-chain interactions that do not spoil the exact solvability. Specifically, the Kitaev honeycomb model can be constructed by this method. So that large-unit-cell brick-wall lattice models constructed in this paper can be viewed as multi-layer models in this scenario, which is in excellent agreement with Kitaev’s argument.

3-Can the authors comment if the models constructed are also solvable using Kitaev's fermionization procedure, or is the Jordan-Wigner construction required?

Re: In principle the spin-1/2 models that we constructed can be exactly solved by using Kitaev’s four-Majorana-fermion decomposition.

However, it is not easy to implement in practice, in particular, when the multiple-spin interactions are taken into consideration, which are essential to lift the local degeneracy (see Ref. 31 for details) in |\nu|\geq2 models. Because, an n-spin term will be mapped to a 2n-Majorana-fermion term in the Kitaev’s fermionization procedure. The searching for constants of motion, i.e. Z_2 static gauge fields, is far from obvious.

In contrast, a systematic way to determine the constants of motion has been established with the help of the Jordan-Wigner formulation (see in Ref. 31 as well).

\nu=2\Gamma

Requested changes

1-I highly recommend the authors add a few more illustrative figures concerning their construction. In particular to visualize the definition of the Jordan-Wigner path that is defined below page 2.

Re: We thank the Referee for this comment. We have sketched the Figure to show the Jordan-Wigner path lacing the 2D lattice. In Figure 1 (c), we have shown the ordering of each site that defines a 1D Jordan-Wigner path.

2-The comment on page 6 "it is natural to make copies of the lattice gauge theory" is confusing and does not actually match the authors construction, as it suggests that there are multiple copies of the Z2 gauge field. Instead, the construction presented only has one Z2 gauge theory (but it is interesting to ask if the model at hand can be recovered from literally taking copies of the model, see my question 2 above).

Re: We thank the Referee for this comment. We have modify the statement to point out that only itinerant Majorana fermions have multiple copies.

Action: We modify the statement in the beginning of section 3.

“it is natural to make \nu copies of itinerant Majorana fermions that are all coupled to one static Z_2 gauge field on a brick-wall-type lattice.”

3-There are a few typos throughout the text, and undefined references on page 6 ("citeTu") and page 7 ("citeMiao1").

Re: We have corrected these typos.

---

## Round 2 · Referee Report · Anonymous (Referee 2) · 2022-11-16

Report

This theoretical paper proposes exactly solvable 2D spin ½ models that can realize any topological order of the Kitaev type (i.e. characterized by a Chern number nu). These model are solved by using a Jordan-Wigner transformation and then a mapping to Majorana fermions (some become itinerant and some realize a static Z_2 gauge field).

I find this paper very interesting with an ingenious (but a bit artificial) construction. It avoids using large spins (e.g. spin-3/2 or related Gamma matrix models) but requires multi-spin interactions (for example, for nu=2q-1, one needs q+1 spin terms) or long-range spin interactions. I think this work should be published. I have a few questions and comments for the authors.

In the first paragraph of the introduction, it is written that “Kitaev proposed a systematic method, dubbed sixteen-fold way, to characterize and classify topological orders in two dimensional quantum systems”. The authors should be more precise and add that they refer to “some” and not “all” 2D topological orders. Indeed, there are many more 2D topological orders then the sixteen ones discussed by Kitaev. According to Wen, any unitary modular tensor category (i.e. S and T matrices, plus chiral central charge c) characterizes a topological order or anyon model. Here the authors are referring to a subset of topological orders, namely those that belong to the D class of the ten-fold periodic table that are characterized by a Chern number. For example, Fibonacci anyons are not one of the 16 topological orders discussed here.

It would be good to enlarge the comparison with previous works trying to obtain all sixteen Kitaev topological orders (Refs 36-39). Such a comparison is done in some detail with respect to Ref 39 on page 6 but it could be done with others refs as well. For example, compared to Ref. 38, the novelty of the present work is in being able to propose an exactly solvable spin-1/2 model that has any Chern number in the vortex free sector (that's the key point), which is supposed to host the ground state at small time-reversal symmetry breaking perturbation.

Is it obvious that the ground-state is in the vortex-free sector? Lieb’s theorem assumes that the graph is bipartite which is not generally the case in your model. Do I understand correctly that pairing between Majorana fermions only occur between black and white sites, so that each specie of Majorana fermion is actually decoupled from the other and hopping on its own brick-wall (odd Chern) or square (even Chern) lattice? If this is true, as brick-wall and square lattices are bipartite, then I understand that, at small time-reversal symmetry breaking perturbation, the ground-state for each specie will be found in the vortex-free sector. But it would help the reader to explain/write that.

End of page 10, “Thus, we are able to realize nu=-2q+1,-2q+2,…,2q-2,2q-1 topological orders…” gives the impression that odd and even Chern numbers are realizable in the same model??? Also, if some species of the Majorana fermions remain gapless, then the complete energy spectrum is not gapped ans there is no meaning in giving a Chern number?

On page 18, you should cite Ref. 18 for the fact that the PBC introduces extra terms in the Hamiltonian involving the fermion parity operator.

What is special with nu mod 16 in your work? I have the impression that you could realize any integer nu.

On page 6, it is stated that “Kitaev honeycomb model is able to host a topological order up to |nu|=1.” This is only true in the vortex free sector. When the time-reversal symmetry breaking term is large enough, the ground-state is in the vortex full sector, which has |nu|=2 (see Lahtinen and Pachos, PRB 2010). In other sectors of the originalk Kitaev model with vortex lattices (which do not host the ground state), one can reach higher values of |nu| such as 3 and 5 (see Ref. 38).

I don’t understand the crucial statement on page 19 that “the pairing term vanishes at k_1 = k_2 = 0”. Which pairing term?

In the comparison with the spin-3/2 YZK model, there are a few points which took me time to understand and should be spelled out by the authors. First, in the even case, each specie of Majorana fermions hops on a square lattice (not a brick-wall). And the YZK model is on a square lattice. Second, the mapping to the YZK model is in the absence of time-reversal symmetry breaking term so that the spectrum is gapless and there is no Chern number (I got confused by the fact that the Hamiltonian in equation 25 and also above equation 24 are labelled by nu=2).

Requested changes

Below equation (6): “IN the gapless phase, a time-reversal-symmetry-breaking perturbation…”
Page 6 “of itinerant Majorana fermions that are coupleD TO a static $Z_2$…”
Page 6 “illustrated by the Kitaev honeycOmb model itself”.
Page 6, Ref. CiteTu
Page 7, Ref. CiteMiao
Page 10, DIRAC cones (instead of Driac cones)
Page 19, “mapped to quAdratic FORM of Majorana fermions”
Page 21, missing w in AcknoWledgement
Strange symbol in the title of ref 27
What are the “Apostrophes” referred to in the caption of Figures 3, 5 and 7?
It would help the reader that the coupling constants in the original Kitaev model recalled in section 2 are also called J_1, J_2 and J_3 instead of J_x, J_y and J_z.

  • validity: high
  • significance: good
  • originality: good
  • clarity: good
  • formatting: -
  • grammar: reasonable

Author:  Yi Zhou  on 2022-12-15  [id 3138]

(in reply to Report 2 on 2022-11-16)

Report
This theoretical paper proposes exactly solvable 2D spin ½ models that can realize any topological order of the Kitaev type (i.e. characterized by a Chern number nu). These model are solved by using a Jordan-Wigner transformation and then a mapping to Majorana fermions (some become itinerant and some realize a static Z_2 gauge field).

I find this paper very interesting with an ingenious (but a bit artificial) construction. It avoids using large spins (e.g. spin-3/2 or related Gamma matrix models) but requires multi-spin interactions (for example, for nu=2q-1, one needs q+1 spin terms) or long-range spin interactions. I think this work should be published. I have a few questions and comments for the authors.

Re: We are grateful for the referee’s constructive comments and suggestions. We have considered all the constructive suggestions made by the referee, enclosed please find our point-to-point response.

In the first paragraph of the introduction, it is written that “Kitaev proposed a systematic method, dubbed sixteen-fold way, to characterize and classify topological orders in two dimensional quantum systems”. The authors should be more precise and add that they refer to “some” and not “all” 2D topological orders. Indeed, there are many more 2D topological orders then the sixteen ones discussed by Kitaev. According to Wen, any unitary modular tensor category (i.e. S and T matrices, plus chiral central charge c) characterizes a topological order or anyon model. Here the authors are referring to a subset of topological orders, namely those that belong to the D class of the ten-fold periodic table that are characterized by a Chern number. For example, Fibonacci anyons are not one of the 16 topological orders discussed here.

Re: We gratefully thank the Referee for pointing out this issue and have changed the statement.

Action: We have modified this sentence as follows: “In a pioneer work [5], Kitaev proposed a systematic method, dubbed “sixteen-fold way", to characterize and classify topological orders in two dimensional (2D) quantum systems that consist of weakly interacting fermions.”

It would be good to enlarge the comparison with previous works trying to obtain all sixteen Kitaev topological orders (Refs 36-39). Such a comparison is done in some detail with respect to Ref 39 on page 6 but it could be done with others refs as well. For example, compared to Ref. 38, the novelty of the present work is in being able to propose an exactly solvable spin-1/2 model that has any Chern number in the vortex free sector (that's the key point), which is supposed to host the ground state at small time-reversal symmetry breaking perturbation.

Re: We thank the Referee for this suggestion.

Action: We have added a Table in the “conclusions and discussion” section to make comparison with previous works, which includes the following items: lattice types, Chern numbers, vortex sectors, and local degrees of freedom.

Is it obvious that the ground-state is in the vortex-free sector? Lieb’s theorem assumes that the graph is bipartite which is not generally the case in your model. Do I understand correctly that pairing between Majorana fermions only occur between black and white sites, so that each species of Majorana fermion is actually decoupled from the other and hopping on its own brick-wall (odd Chern) or square (even Chern) lattice? If this is true, as brick-wall and square lattices are bipartite, then I understand that, at small time-reversal symmetry breaking perturbation, the ground-state for each specie will be found in the vortex-free sector. But it would help the reader to explain/write that.

Re: This is indeed true as understood by the Referee. The way that we determine the ground state flux sector has already been stated in our paper. The relevant statements can be found in the following:

(i) above Eqs. (10): “Moreover, the pairing of each species of itinerant Majorana fermions is of the same form as that in Eq. (4). Therefore, the ground state of Hodd must be a zero flux state on which all the Z2 fluxes \phi_{p,\mathbit{r}}^{odd}=1.”
(ii) above Eq. (16): “Besides, the Hamiltonian for each species of the itinerant Majorana fermions is equivalent to a Hamiltonian defined on a square lattice whose unit cell consists of two sites (r,A,μ) and (r, B,μ). Thus the Lieb’s theorem [41] for square lattice is applicable to Heven as well: the ground state of Heven is in the flux sector where all the eigenvalues of \phi_{p,\mathbit{r}}^{even}=\phi_{p\prime,\mathbit{r}}^{even}=-1 (π-flux sector).”
(iii) above Eqs.(21): “Since the pairing form of each species of itinerant Majorana fermions takes the same form as that in Eq. (4), the ground state of\ {\widetilde{H}}^{even}
must be in the zero flux sector on which {\widetilde{\phi}}_{p,\mathbit{r}}^{even}=1 everywhere.”

End of page 10, “Thus, we are able to realize nu=-2q+1,-2q+2,…,2q-2,2q-1 topological orders…” gives the impression that odd and even Chern numbers are realizable in the same model??? Also, if some species of the Majorana fermions remain gapless, then the complete energy spectrum is not gapped and there is no meaning in giving a Chern number?

Re: We have rewritten the last paragraph in section 3.1 to avoid possible misunderstand.

Action: The last paragraph in section 3.1 has been written as follows:

“… such that some of the 2q − 1 species of Majorana fermions remain gapless with Dirac cones, meanwhile other species are fully gapped by H^{odd} itself, i.e. in the Abelian phase [5]. A small time-reversal symmetry breaking perturbation can open an energy gap at each Dirac cone, resulting in a non-Abelian state, on which each species of Majorana fermions gives rise to a Chern number ν = ±1.”

On page 18, you should cite Ref. 18 for the fact that the PBC introduces extra terms in the Hamiltonian involving the fermion parity operator.

Action: Thank you. This citation has been done.

What is special with nu mod 16 in your work? I have the impression that you could realize any integer nu.

Re: We thank the referee for this question. Indeed, any integer \nu can be realized in our models.\nu However, as pointed out by Kitaev in his original paper, if one considers the topological aspect only, i.e., the anyon statistics, in the models that consist of free fermions coupled to a Z2 gauge field, these models can be classified in terms of the topological spin of Z2 vortices, \theta_\sigma=e^{i\pi\nu/8}. Since it determines the braiding and fusion rules of anyons up to a trivial global phase factor.

Certainly, the strong interactions between fermions or additionally imposed symmetries can modify this sixteen-fold way classification. In our model, the Z2 gauge field is static, so that the interaction between fermions is weak and will not modify the sixteen-fold way. However, there remains some room to impose additional symmetries to the coupling constants J’s, such that the sixteen topological orders can be enriched by these additional symmetries.=\nu\ mod\ 16

On page 6, it is stated that “Kitaev honeycomb model is able to host a topological order up to |nu|=1.” This is only true in the vortex free sector. When the time-reversal symmetry breaking term is large enough, the ground-state is in the vortex full sector, which has |nu|=2 (see Lahtinen and Pachos, PRB 2010). In other sectors of the original Kitaev model with vortex lattices (which do not host the ground state), one can reach higher values of |nu| such as 3 and 5 (see Ref. 38).

Re: We thank the Referee for pointing out this issue. We have added a new reference [42] (Lahtinen and Pachos, PRB 2010) and a footnote on page 5 to clarify it.

Action: A footnote has been added in the end of page 5, and the reference [42] has been cited.

“Here a small time-reversal symmetry breaking perturbation can be added to open an energy gap. It is also worth mentioning that for large time-reversal symmetry breaking term, the ground-state is in the vortex full sector has a Chern number ν = 2 [42].”\nu=1\nu=2

I don’t understand the crucial statement on page 19 that “the pairing term vanishes at k_1 = k_2 = 0”. Which pairing term?

Re: We thank the referee for this question, and have rewritten the last part in section 4.2 for clarification.

Action: The last paragraph in section 4.2 has been modified, where we have discussed how the p+ip pairing term arises in a topologically non-trivial band.

In the comparison with the spin-3/2 YZK model, there are a few points which took me time to understand and should be spelled out by the authors. First, in the even case, each species of Majorana fermions hops on a square lattice (not a brick-wall). And the YZK model is on a square lattice. Second, the mapping to the YZK model is in the absence of time-reversal symmetry breaking term so that the spectrum is gapless and there is no Chern number (I got confused by the fact that the Hamiltonian in equation 25 and also above equation 24 are labelled by nu=2).

Re: We thank the referee for raising this issue. First, the brick-wall lattice can be transformed into a square lattice by a coarse-graining scheme, as shown in Fig. 9 in this revised version.

Second, the YZK model itself is gapless when the time-reversal symmetry (TRS) is respected, just as our \nu=2 model and the original Kitaev model (the \nu=1 or gapless phase). However, a small TRS breaking term will open an energy gap in the gapless state and give rise to a finite Chern number \nu. Meanwhile, the absolute value of \nu is determined by the helicity of the Dirac cones and will not be affected by the detail of the TRS breaking perturbation. So that we follow the convention in the literature to use the notation \nu=2 here.

Action: We add a new figure (Fig. 9 in this revised version) to illustrate the coarse-graining scheme.

\nu=2\nu=2Requested changes
Below equation (6): “IN the gapless phase, a time-reversal-symmetry-breaking perturbation…”
Page 6 “of itinerant Majorana fermions that are coupleD TO a static …”
Page 6 “illustrated by the Kitaev honeycOmb model itself”.
Page 6, Ref. CiteTu
Page 7, Ref. CiteMiao
Page 10, DIRAC cones (instead of Driac cones)
Page 19, “mapped to quAdratic FORM of Majorana fermions”
Page 21, missing w in AcknoWledgement
Strange symbol in the title of ref 27
What are the “Apostrophes” referred to in the caption of Figures 3, 5 and 7?
It would help the reader that the coupling constants in the original Kitaev model recalled in section 2 are also called J_1, J_2 and J_3 instead of J_x, J_y and J_z.

Re: We have corrected these typos. The “Apostrophes” refer to the symbols between the spin operators (labeled by x, y, z) in the above figures. We use them to represent \sigma^z on corresponding sites. To avoid confusion, we have modified these figures and changed these symbols into stars in blue.

---

## Round 3 · Referee Report · Anonymous (Referee 2) · 2022-12-26

Report

The authors have taken all remarks into account and made a really nice job in improving their manuscript (in particular the table in the conclusion). I have no further important comment and think that the paper should now be published as it is.

Minor remarks: - In the footnote 1, there is a missing “and”: “….the ground-state is in the vortex full sector AND has a Chern number nu=2.” - The doi is missing in the new ref 42

---

## Round 3 · Referee Report · Anonymous (Referee 1) · 2023-1-19

Report

The authors have responded to most of my criticisms and questions in a satisfactory manner, and have made some more improvements beyond that. In particular, the overview over various constructions of microscopical models will be helpful for future reference.

I stand by my criticism that this model is rather artificial, and the authors have not provided any details concerning any particular quantum simulation/computing platform that may have a particularly suitable architecture for this type of model. However, I can see merit in having an explicit S=1/2 model at hand, which may allow for further studies into explicit realisations on qubit architectures. I therefore recommend the paper for publication.

---

## Round 3 · Author Response

Dear the Editor,

Many thanks for handling our submission and sending us the referee reports.

We thank the referees for their positive evaluation on our work and constructive comments which really improve our manuscript.

Having convincingly addressed all the concerns of the referees, and having revised our manuscript accordingly, we believe this improved version is ready for publication in SciPost Physics.

Yours sincerely, Jin-Tao Jin, Jian-Jian Miao, and Yi Zhou

Point-to-point response to Referees’ comments/suggestions:

Reviewer(s)' Comments to Author:

Reviewer: 1

Report This theoretical paper proposes exactly solvable 2D spin ½ models that can realize any topological order of the Kitaev type (i.e. characterized by a Chern number nu). These model are solved by using a Jordan-Wigner transformation and then a mapping to Majorana fermions (some become itinerant and some realize a static Z_2 gauge field).

I find this paper very interesting with an ingenious (but a bit artificial) construction. It avoids using large spins (e.g. spin-3/2 or related Gamma matrix models) but requires multi-spin interactions (for example, for nu=2q-1, one needs q+1 spin terms) or long-range spin interactions. I think this work should be published. I have a few questions and comments for the authors.

Re: We are grateful for the referee’s constructive comments and suggestions. We have considered all the constructive suggestions made by the referee, enclosed please find our point-to-point response.

In the first paragraph of the introduction, it is written that “Kitaev proposed a systematic method, dubbed sixteen-fold way, to characterize and classify topological orders in two dimensional quantum systems”. The authors should be more precise and add that they refer to “some” and not “all” 2D topological orders. Indeed, there are many more 2D topological orders then the sixteen ones discussed by Kitaev. According to Wen, any unitary modular tensor category (i.e. S and T matrices, plus chiral central charge c) characterizes a topological order or anyon model. Here the authors are referring to a subset of topological orders, namely those that belong to the D class of the ten-fold periodic table that are characterized by a Chern number. For example, Fibonacci anyons are not one of the 16 topological orders discussed here.

Re: We gratefully thank the Referee for pointing out this issue and have changed the statement.

Action: We have modified this sentence as follows: “In a pioneer work [5], Kitaev proposed a systematic method, dubbed “sixteen-fold way", to characterize and classify topological orders in two dimensional (2D) quantum systems that consist of weakly interacting fermions.”

It would be good to enlarge the comparison with previous works trying to obtain all sixteen Kitaev topological orders (Refs 36-39). Such a comparison is done in some detail with respect to Ref 39 on page 6 but it could be done with others refs as well. For example, compared to Ref. 38, the novelty of the present work is in being able to propose an exactly solvable spin-1/2 model that has any Chern number in the vortex free sector (that's the key point), which is supposed to host the ground state at small time-reversal symmetry breaking perturbation.

Re: We thank the Referee for this suggestion.

Action: We have added a Table in the “conclusions and discussion” section to make comparison with previous works, which includes the following items: lattice types, Chern numbers, vortex sectors, and local degrees of freedom.

Is it obvious that the ground-state is in the vortex-free sector? Lieb’s theorem assumes that the graph is bipartite which is not generally the case in your model. Do I understand correctly that pairing between Majorana fermions only occur between black and white sites, so that each species of Majorana fermion is actually decoupled from the other and hopping on its own brick-wall (odd Chern) or square (even Chern) lattice? If this is true, as brick-wall and square lattices are bipartite, then I understand that, at small time-reversal symmetry breaking perturbation, the ground-state for each specie will be found in the vortex-free sector. But it would help the reader to explain/write that.

Re: This is indeed true as understood by the Referee. The way that we determine the ground state flux sector has already been stated in our paper. The relevant statements can be found in the following:

(i) above Eqs. (10): “Moreover, the pairing of each species of itinerant Majorana fermions is of the same form as that in Eq. (4). Therefore, the ground state of Hodd must be a zero flux state on which all the Z2 fluxes \phi_{p,\mathbit{r}}^{odd}=1.” (ii) above Eq. (16): “Besides, the Hamiltonian for each species of the itinerant Majorana fermions is equivalent to a Hamiltonian defined on a square lattice whose unit cell consists of two sites (r,A,μ) and (r, B,μ). Thus the Lieb’s theorem [41] for square lattice is applicable to Heven as well: the ground state of Heven is in the flux sector where all the eigenvalues of \phi_{p,\mathbit{r}}^{even}=\phi_{p\prime,\mathbit{r}}^{even}=-1 (π-flux sector).” (iii) above Eqs.(21): “Since the pairing form of each species of itinerant Majorana fermions takes the same form as that in Eq. (4), the ground state of\ {\widetilde{H}}^{even} must be in the zero flux sector on which {\widetilde{\phi}}_{p,\mathbit{r}}^{even}=1 everywhere.”

End of page 10, “Thus, we are able to realize nu=-2q+1,-2q+2,…,2q-2,2q-1 topological orders…” gives the impression that odd and even Chern numbers are realizable in the same model??? Also, if some species of the Majorana fermions remain gapless, then the complete energy spectrum is not gapped and there is no meaning in giving a Chern number?

Re: We have rewritten the last paragraph in section 3.1 to avoid possible misunderstand.

Action: The last paragraph in section 3.1 has been written as follows:

“… such that some of the 2q − 1 species of Majorana fermions remain gapless with Dirac cones, meanwhile other species are fully gapped by H^{odd} itself, i.e. in the Abelian phase [5]. A small time-reversal symmetry breaking perturbation can open an energy gap at each Dirac cone, resulting in a non-Abelian state, on which each species of Majorana fermions gives rise to a Chern number ν = ±1.”

On page 18, you should cite Ref. 18 for the fact that the PBC introduces extra terms in the Hamiltonian involving the fermion parity operator.

Action: Thank you. This citation has been done.

What is special with nu mod 16 in your work? I have the impression that you could realize any integer nu.

Re: We thank the referee for this question. Indeed, any integer \nu can be realized in our models.\nu However, as pointed out by Kitaev in his original paper, if one considers the topological aspect only, i.e., the anyon statistics, in the models that consist of free fermions coupled to a Z2 gauge field, these models can be classified in terms of the topological spin of Z2 vortices, \theta_\sigma=e^{i\pi\nu/8}. Since it determines the braiding and fusion rules of anyons up to a trivial global phase factor.

Certainly, the strong interactions between fermions or additionally imposed symmetries can modify this sixteen-fold way classification. In our model, the Z2 gauge field is static, so that the interaction between fermions is weak and will not modify the sixteen-fold way. However, there remains some room to impose additional symmetries to the coupling constants J’s, such that the sixteen topological orders can be enriched by these additional symmetries.=\nu\ mod\ 16

On page 6, it is stated that “Kitaev honeycomb model is able to host a topological order up to |nu|=1.” This is only true in the vortex free sector. When the time-reversal symmetry breaking term is large enough, the ground-state is in the vortex full sector, which has |nu|=2 (see Lahtinen and Pachos, PRB 2010). In other sectors of the original Kitaev model with vortex lattices (which do not host the ground state), one can reach higher values of |nu| such as 3 and 5 (see Ref. 38).

Re: We thank the Referee for pointing out this issue. We have added a new reference [42] (Lahtinen and Pachos, PRB 2010) and a footnote on page 5 to clarify it.

Action: A footnote has been added in the end of page 5, and the reference [42] has been cited.

“Here a small time-reversal symmetry breaking perturbation can be added to open an energy gap. It is also worth mentioning that for large time-reversal symmetry breaking term, the ground-state is in the vortex full sector has a Chern number ν = 2 [42].”\nu=1\nu=2

I don’t understand the crucial statement on page 19 that “the pairing term vanishes at k_1 = k_2 = 0”. Which pairing term?

Re: We thank the referee for this question, and have rewritten the last part in section 4.2 for clarification.

Action: The last paragraph in section 4.2 has been modified, where we have discussed how the p+ip pairing term arises in a topologically non-trivial band.

In the comparison with the spin-3/2 YZK model, there are a few points which took me time to understand and should be spelled out by the authors. First, in the even case, each species of Majorana fermions hops on a square lattice (not a brick-wall). And the YZK model is on a square lattice. Second, the mapping to the YZK model is in the absence of time-reversal symmetry breaking term so that the spectrum is gapless and there is no Chern number (I got confused by the fact that the Hamiltonian in equation 25 and also above equation 24 are labelled by nu=2).

Re: We thank the referee for raising this issue. First, the brick-wall lattice can be transformed into a square lattice by a coarse-graining scheme, as shown in Fig. 9 in this revised version.

Second, the YZK model itself is gapless when the time-reversal symmetry (TRS) is respected, just as our \nu=2 model and the original Kitaev model (the \nu=1 or gapless phase). However, a small TRS breaking term will open an energy gap in the gapless state and give rise to a finite Chern number \nu. Meanwhile, the absolute value of \nu is determined by the helicity of the Dirac cones and will not be affected by the detail of the TRS breaking perturbation. So that we follow the convention in the literature to use the notation \nu=2 here.

Action: We add a new figure (Fig. 9 in this revised version) to illustrate the coarse-graining scheme.

\nu=2\nu=2Requested changes Below equation (6): “IN the gapless phase, a time-reversal-symmetry-breaking perturbation…” Page 6 “of itinerant Majorana fermions that are coupleD TO a static …” Page 6 “illustrated by the Kitaev honeycOmb model itself”. Page 6, Ref. CiteTu Page 7, Ref. CiteMiao Page 10, DIRAC cones (instead of Driac cones) Page 19, “mapped to quAdratic FORM of Majorana fermions” Page 21, missing w in AcknoWledgement Strange symbol in the title of ref 27 What are the “Apostrophes” referred to in the caption of Figures 3, 5 and 7? It would help the reader that the coupling constants in the original Kitaev model recalled in section 2 are also called J_1, J_2 and J_3 instead of J_x, J_y and J_z.

Re: We have corrected these typos. The “Apostrophes” refer to the symbols between the spin operators (labeled by x, y, z) in the above figures. We use them to represent \sigma^z on corresponding sites. To avoid confusion, we have modified these figures and changed these symbols into stars in blue.

Reviewer: 2

Strengths 1- A new construction of Kitaev-inspired lattice models which realize the 16-fold way of topological orders 2- The models constructed have elementary degrees of freedom and might thus be implemented in quantum simulators (cold-atom experiments) more easily 3- The authors provide a great amount of technical detail, making calculations reproducible

Weaknesses 1- The paper will be challenging to read for non-experts (as it is heavy on technical details) and lacks illustrations of some key concepts

Re: We have revised the paper in accordance with the reviewer’s suggestions. Hope this revised version will be more reader friendly.

2-Lacking details on realization: It is not clear to what extent the models are actually more "realistic" than previous constructions and how they can be realized in quantum simulators/quantum computers

Re: To be honest, we are not experts on quantum simulators/quantum computers. However, as a common sense, we believe that local two-state devices, i.e., qubits, are much easier to realize, in comparison with four (or more)-state devices in Josephson junctions or other quantum computation platforms.

Report The paper is concerned with constructing bosonic microscopic lattice models which realize the 16-fold way of topological orders classified by Kitaev, which are obtained by coupling (gapped) Majorana fermions with Chern number to a Z2 gauge theory. Previous attempts have focused on generating the required Majorana bands by flux engineering, or have started with enlarged local Hilbert spaces. In contrast, the authors of the work at hand are motivated by the Jordan-Wigner solution of Kitaev's honeycomb model and construct generalized brick-wall lattices of S=1/2 degrees of freedom. They appropriately gap out the itinerant Majorana fermions and demonstrate the topological degeneracies of the even/odd phases on the torus.

The work seems valid and is of value as it explores a new way of constructing microscopic models for the 16fold way. The paper is mostly well written and provides a lot technical details. However, I am not (yet) convinced the paper deserves publication in SciPost Physics given the acceptance criteria. Microscopic models for the 16fold way have been presented before so the work does not qualify as a breakthrough, and despite the author's comments, it appears that the model is not straightforwardly studied in quantum simulators (I wonder if the authors had any specific platforms/details concerning an AMO-implementation in mind?)

Re: We are grateful for the referee’s questions. We have considered all the constructive comments made by the referee, enclosed please find our point-to-point response.

I also have two more detailed conceptual questions that the authors could address in their work:

1-The authors present a neat mapping which shows that the Yao-Zhang-Kivelson can be recovered within their construction. Indeed, I wonder if this can be generalized to all \Gamma-matrix models. It seems to me that grouping a sufficient number of S=1/2 will effectively give rise to one site with a larger Hilbert space on which the \Gamma-matrices act?

Re: We thank the Referee for this question. Certainly, one can grouping S=1/2 spins to construct \Gamma-matrix models, in analogy with the coarse-graining scheme illustrated in Fig. 9 in this revised version. However, such a coarse-graining procedure is not necessary to be invertible. Therefore, the question whether an exactly solvable \Gamma-matrix model can be mapped to a spin-1/2 model is still open, unless an explicit invert-construction has been done as we did for the Yao-Zhang-Kivelson model.

\times\Gamma\Gamma\times\Gamma\Gamma\Gamma\Gamma\ \Gamma\ \Gamma 2-In the construction at hand, we have multiple Majorana fermion bands coupled to one Z2 gauge field. Kitaev's proposal for constructing higher-orders starts with multiple layers and then condenses interlayer anyons (see Sec. 10.9 in https://arxiv.org/abs/cond-mat/0506438v3) which binds the gauge fields together. It seems that the lacing construction presented by the authors may also be understood as the result of some layered construction. Could the authors comment on possible relations?

Re: Indeed, it has been shown in Ref. 31 that exactly solvable spin-1/2 models can be constructed in arbitrary dimensions by using the lacing method systematically. The basic idea is that a brick-wall lattice model can be made by coupling exactly solvable spin-1/2 chains via inter-chain interactions that do not spoil the exact solvability. Specifically, the Kitaev honeycomb model can be constructed by this method. So that large-unit-cell brick-wall lattice models constructed in this paper can be viewed as multi-layer models in this scenario, which is in excellent agreement with Kitaev’s argument.

3-Can the authors comment if the models constructed are also solvable using Kitaev's fermionization procedure, or is the Jordan-Wigner construction required?

Re: In principle the spin-1/2 models that we constructed can be exactly solved by using Kitaev’s four-Majorana-fermion decomposition.

However, it is not easy to implement in practice, in particular, when the multiple-spin interactions are taken into consideration, which are essential to lift the local degeneracy (see Ref. 31 for details) in |\nu|\geq2 models. Because, an n-spin term will be mapped to a 2n-Majorana-fermion term in the Kitaev’s fermionization procedure. The searching for constants of motion, i.e. Z_2 static gauge fields, is far from obvious.

In contrast, a systematic way to determine the constants of motion has been established with the help of the Jordan-Wigner formulation (see in Ref. 31 as well).

\nu=2\Gamma

Requested changes

1-I highly recommend the authors add a few more illustrative figures concerning their construction. In particular to visualize the definition of the Jordan-Wigner path that is defined below page 2.

Re: We thank the Referee for this comment. We have sketched the Figure to show the Jordan-Wigner path lacing the 2D lattice. In Figure 1 (c), we have shown the ordering of each site that defines a 1D Jordan-Wigner path.

2-The comment on page 6 "it is natural to make copies of the lattice gauge theory" is confusing and does not actually match the authors construction, as it suggests that there are multiple copies of the Z2 gauge field. Instead, the construction presented only has one Z2 gauge theory (but it is interesting to ask if the model at hand can be recovered from literally taking copies of the model, see my question 2 above).

Re: We thank the Referee for this comment. We have modify the statement to point out that only itinerant Majorana fermions have multiple copies.

Action: We modify the statement in the beginning of section 3.

“it is natural to make \nu copies of itinerant Majorana fermions that are all coupled to one static Z_2 gauge field on a brick-wall-type lattice.”

3-There are a few typos throughout the text, and undefined references on page 6 ("citeTu") and page 7 ("citeMiao1").

Re: We have corrected these typos.

---

## Round 3 · List of Changes

Summary of changes (the changes are marked in blue in the main text):

(1) Modify the second sentence in the first paragraph of the introduction as follows:

“In a pioneer work [5], Kitaev proposed a systematic method, dubbed “sixteen-fold way", to characterize and classify topological orders in two dimensional (2D) quantum systems that consist of weakly interacting fermions.”

(2) Add a Table to the “Conclusions and discussions” section.

(3) Modify the statement at the end of section 3.1 as follows:

“… such that some of the 2q − 1 species of Majorana fermions remain gapless with Dirac cones, meanwhile other species are fully gapped by H^{odd} itself, i.e. in the Abelian phase [5]. A small time-reversal symmetry breaking perturbation can open an energy gap at each Dirac cone, resulting in a non-Abelian state, on which each species of Majorana fermions gives rise to a Chern number ν = ±1.”

(4) Add a footnote in the end of page 5 and a reference [42]:

“\nu=1\nu=2Here a small time-reversal symmetry breaking perturbation can be added to open an energy gap. It is also worth mentioning that for large time-reversal symmetry breaking term, the ground-state is in the vortex full sector has a Chern number ν = 2 [42].”

(5) Modify the second sentence in the second paragraph as follows:

“… it is natural to make \nu copies of itinerant Majorana fermions that are all coupled to a single static Z_2 gauge field on a brick-wall-type lattice.”

(6) The last paragraph in section 4.2 has been modified, where we have discussed how the p+ip pairing term arises in a topologically non-trivial band.

(7) Add a new figure (Fig. 9 in this revised version) to illustrate the coarse-graining scheme. Modify the statement in the second paragraph of “(i)” in “Conclusions and discussions” section as follows:

“Indeed, this equivalence can be understood as follows: As shown in Fig. 9, the q = 2 version of brick-wall lattice in Fig. 5 (a) can be transformed into a square lattice via coarse-graining [see Fig. 9 (b)].”

(8) Correct the typos including those required by the reviewers. Correct figures and captions on request by the reviewers.

---

## Editorial Decision

published